# VEQ: Modality-Adaptive Quantization for MoE Vision-Language Models

Guangshuo Qin [* 1]  Zhiteng Li [* 1]  Zheng Chen [1]  Weihang Zhang [2]  Linghe Kong [1]  Yulun Zhang [† 1]

## Abstract

Mixture-of-Experts(MoE) Vision-Language Models (VLMs) offer remarkable performance but incur prohibitive memory and computational costs, making compression essential. Post-Training Quantization (PTQ) is an effective training-free technique to address the massive memory and computation overhead. Existing quantization paradigms fall short as they are oblivious to two critical forms of heterogeneity: the inherent discrepancy between vision and language tokens, and the non-uniform contribution of different experts. To bridge this gap, we propose Visual Expert Quantization (VEQ), a dual-aware quantization framework designed to simultaneously accommodate cross-modal differences and heterogeneity between experts. Specifically, VEQ incorporates 1)**Modality-expert-aware Quantization**, which utilizes expert activation frequency to prioritize error minimization for pivotal experts, and 2)**Modality-affinity-aware Quantization**, which constructs an enhanced Hessian matrix by integrating token-expert affinity with modality information to guide the calibration process. Extensive experiments across diverse benchmarks verify that VEQ consistently outperforms state-of-the-art baselines. Specifically, under the W3A16 configuration, our method achieves significant average accuracy gains of 2.04% on Kimi-VL and 3.09% on Qwen3-VL compared to the previous SOTA quantization methods, demonstrating superior robustness across various multimodal tasks. Our code will be available at https://github.com/guangshuoqin/VEQ.

## 1. Introduction

In recent years, Vision-Language Models (VLMs) have demonstrated unprecedented proficiency across a broad

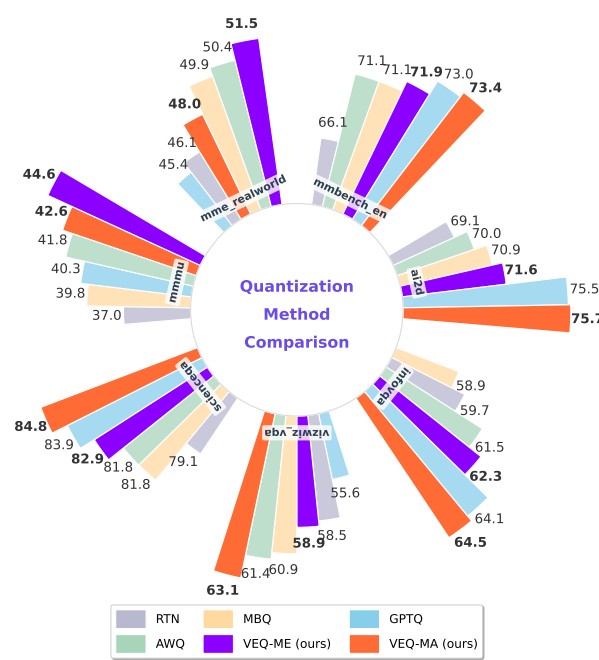

*Figure 1.* Zero-shot performance of Kimi-VL-Instruct under 3-bit weight quantization (W3A16). Our methods consistently outperform established baselines, demonstrating superior robustness.

spectrum of multimodal tasks, ranging from fundamental visual question answering and image captioning to complex visual reasoning (Radford et al., 2021; Alayrac et al., 2022; Li et al., 2023). By seamlessly aligning visual perception with linguistic semantics, these models effectively bridge the cross-modal divide, empowering intelligent agents to perceive, reason, and interact with the physical environment with human-level versatility (Lu et al., 2019; Hurst et al., 2024). With the increasing demand for robust multimodal understanding, VLMs have become increasingly important for practical applications.

To further scale up model capacity while maintaining computational efficiency, the Mixture-of-Experts (MoE) architecture has been widely adopted in state-of-the-art VLMs. Unlike dense models that activate all parameters for every token, MoE models utilize a sparse routing mechanism to activate only a subset of experts, effectively reducing inference costs while preserving a vast parameter space (Fedus et al., 2022). Prominent open-source VLMs, such as DeepSeek-VL2 (Wu et al., 2024), Kimi-VL (Team et al., 2025), Qwen3-VL (Bai et al., 2025), and ERNIE-4.5-VL

---

*Equal contribution [1]Shanghai Jiao Tong University [2]Tsinghua University. Correspondence to: Yulun Zhang[†] <yulun100@gmail.com>.

*Proceedings of the 43rd International Conference on Machine Learning*, Seoul, South Korea. PMLR 306, 2026. Copyright 2026 by the author(s).

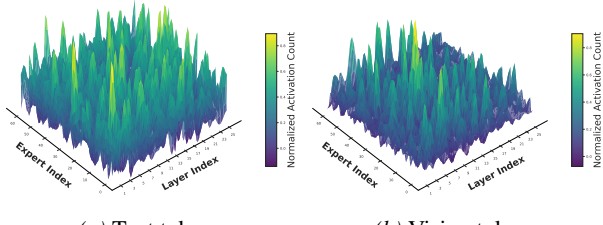

*(a)* Text token.  *(b)* Vision token.

*Figure 2.* Comparative analysis of activation characteristics across different modalities. Peaks represent high activation frequency.

(Baidu-ERNIE-Team, 2025), have successfully leveraged this architecture to achieve superior performance with manageable resource consumption. Despite their efficiency advantages over dense counterparts, MoE VLMs still incur significant memory footprints and latency during inference, necessitating effective model compression techniques. Post-Training Quantization (PTQ) has emerged as a practical solution. Mainstream weight-only quantization methods, such as AWQ (Lin et al., 2024) and GPTQ (Frantar et al., 2022), primarily focus on minimizing weight reconstruction error to preserve model performance. Meanwhile, activation-aware methods like SmoothQuant (Xiao et al., 2024) and SpinQuant (Liu et al., 2024b) address the challenge of activation outliers by smoothing or rotating the feature space. More recently, MBQ (Li et al., 2025) introduced a novel perspective by exploring the sensitivity of multimodal tokens to further minimize quantization error.

However, directly applying these conventional quantization methods to MoE VLMs proves suboptimal. Existing approaches typically treat the model as a monolithic dense structure, thereby neglecting the inherent structural sparsity of MoE architectures. Specifically, they fail to account for the varying importance of different experts. As illustrated in Figure 2, a small subset of hot experts is accessed more frequently and dominates the output, while other experts remain dormant (Fedus et al., 2022; Chi et al., 2022). Furthermore, current methods often neglect the distinct statistical distribution heterogeneity between vision and text tokens (Liang et al., 2022). Applying a unified quantization strategy across these modalities leads to significant performance deterioration, as the sensitivity to quantization noise varies drastically between continuous visual embeddings and discrete textual representations.

To address these challenges, we propose Visual Expert Quantization (**VEQ**), a novel framework tailored for MoE VLMs. We conduct comprehensive evaluations on leading MoE architectures, including Kimi-VL (Team et al., 2025) and Qwen3-VL (Bai et al., 2025), across challenging multimodal benchmarks such as MMMU (Yue et al., 2024), MME-RealWorld (Zhang et al., 2024b), MMBench (Liu et al., 2024a), and InfoVQA (Mathew et al., 2022). Empirical results demonstrate that VEQ consistently outperforms established baselines (e.g., AWQ (Lin et al., 2024), GPTQ

(Frantar et al., 2022), MBQ (Li et al., 2025)), establishing a new state-of-the-art for quantized MoE VLMs. Notably, VEQ exhibits superior robustness in low-bit settings. To the best of our knowledge, this represents a pioneering effort in compressing large-scale MoE VLMs.

Our main contributions are summarized as follows:

- **Pioneering Framework for MoE VLMs:** To the best of our knowledge, this work represents the first attempt to simultaneously address the dual challenges of multi-modal heterogeneity and the unique structural properties of Mixture-of-Experts (MoE) based architectures in the context of quantization.
- **Modality-Expert-Aware Quantization:** We propose a novel strategy that assigns importance scores to experts based on the routing frequency of visual and textual tokens. By explicitly modeling these routing patterns, we utilize the scores to effectively minimize quantization error in critical experts.
- **Modality-Affinity-Aware Quantization:** We introduce a refinement method that leverages router affinity logits and input token modalities to re-weight the Hessian matrix. This approach incorporates semantic affinity into the optimization process, further enhancing the precision of the quantized model.

## 2. Related Work
### 2.1. VLM Quantization
Post-Training Quantization (PTQ) for Vision-Language Models (VLMs) remains a challenging and relatively under-explored area, primarily due to the distribution heterogeneity between vision and text modalities. VLMQ (Xue et al., 2026) addresses visual token redundancy by proposing an importance-aware objective. It generates an enhanced Hessian matrix incorporating token-level importance factors via a lightweight block-wise backward pass. Q-VLM (Wang et al., 2024) utilizes activation entropy as a proxy to identify cross-layer dependencies for efficient block partitioning. It further optimizes the visual encoder to disentangle these dependencies, thereby reducing search overhead while maintaining accuracy. MBQ (Li et al., 2025) accounts for the distinct sensitivity levels of vision and language tokens by incorporating gradient-based sensitivity indicators into the calibration process, aiming to balance reconstruction loss across modalities. Bi-VLM (Wang et al., 2025) implements a saliency-aware hybrid quantization algorithm that partitions weights non-uniformly based on Gaussian quantiles. This approach assigns higher precision to salient outliers while binarizing the remaining parameters. MQuant (Yu et al., 2025) introduces modality-specific static quantization and an attention-invariant switching mechanism to address distribution disparities. Additionally, it employs Rotation Magnitude Suppression (RMS) to mitigate outliers induced by online Hadamard transformations.

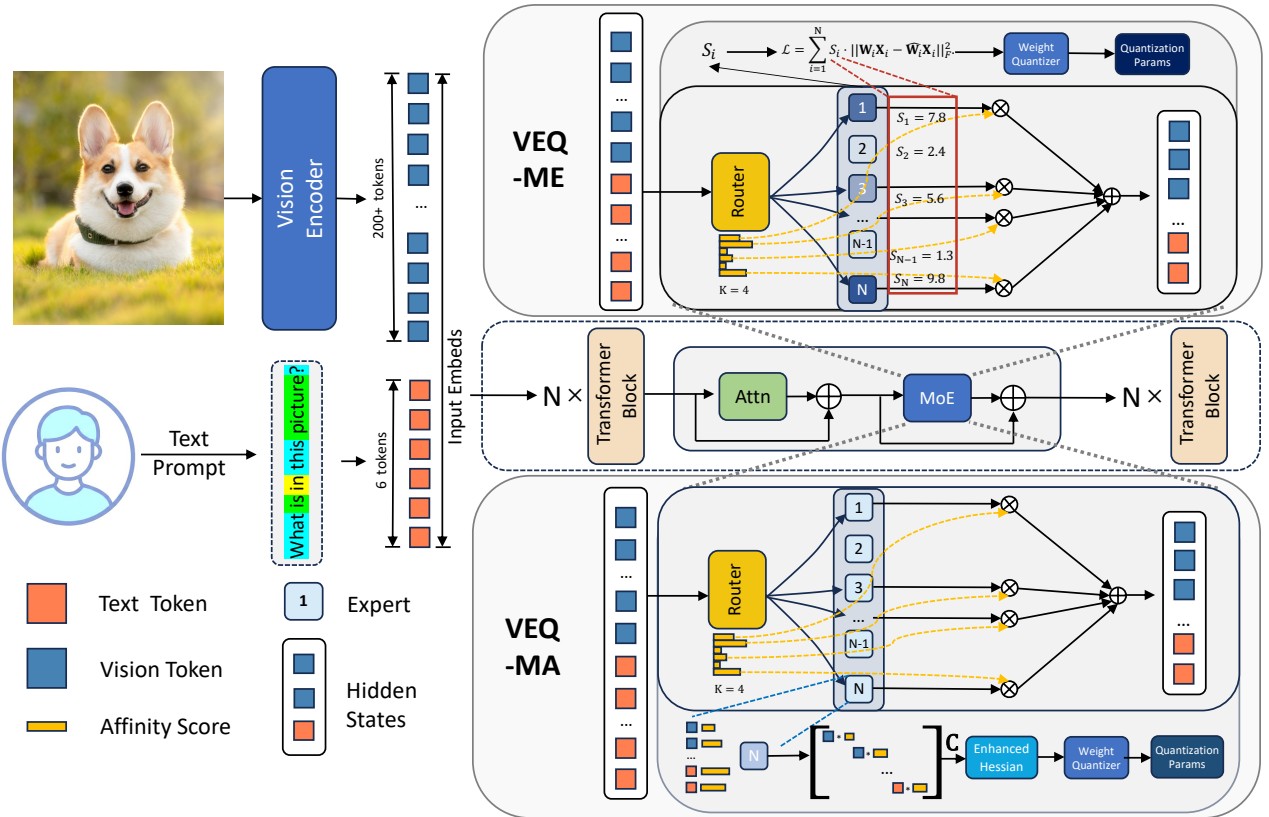

*Figure 3.* Overview of the proposed VEQ framework. Our method consists of two core components: (1) **VEQ-ME**, which dynamically assigns importance scores $S_i$ to experts based on their activation frequencies, thereby prioritizing error minimization for pivotal experts in the reconstruction loss; and (2) **VEQ-MA**, which constructs an enhanced Hessian matrix by integrating token-expert affinity scores and modality sensitivity, enabling the calibration process to adapt to the varying sensitivities of multi-modal tokens.

## 2.2. MoE LLM Quantization

PTQ also presents significant difficulties for Mixture-of-Experts (MoE) architectures, stemming from intrinsic expert sparsity and the complex affinity between tokens and experts. Several recent studies have aimed to address these issues. MoEQuant (Hu et al., 2025) tackles inter- and intra-expert imbalances by employing an expert-balanced self-sampling method for calibration. It further utilizes an affinity-guided quantization strategy that weights errors according to token-expert correlations. MoQa (Zheng et al., 2026) implements an expert-level mixed-precision base quantization via multistage data-model distribution analysis, complemented by a channel-level dynamic adjustment mechanism to adapt to novel data distributions. MoQE (Zhang et al., 2025) leverages the MoE architecture for inference acceleration by treating multiple quantization variants of a single model as experts, using a lightweight router to dynamically assign input data to the optimal quantization expert. MxMoE (Duanmu et al., 2025) proposes an accuracy-performance co-design framework that allocates bit-widths at the granularity of linear blocks. This is achieved by analyzing parameter sensitivity and expert activation frequencies to optimize mixed-precision configurations for quantization.

## 3. Method

In this section, we propose **Visual Expert Quantization (VEQ)**, a novel post-training quantization framework tailored for MoE VLMs. We begin by analyzing the intrinsic heterogeneity of MoE VLM in Section 3.1. Building on these observations, we formulate a modality-aware expert importance metric in Section 3.2. Finally, we introduce an affinity-aware quantization algorithm to minimize reconstruction error in Section 3.3.

### 3.1. Heterogeneity in MoE VLM Quantization

#### 3.1.1. MODALITY HETEROGENEITY

Distinct modalities exhibit unique routing patterns and sensitivity levels within MoE layers. To quantify this heterogeneity, we utilize the gradient of the Supervised Fine-Tuning (SFT) loss as a metric to measure the error sensitivity of vision and text tokens.

Vision tokens, characterized by spatial redundancy, generally demonstrate lower gradient magnitudes, implying a lower sensitivity regarding their impact on inference results. In contrast, information-dense text tokens dominate the output distribution. As illustrated in Figure 4, Our analysis of 128 samples from the COCO (Lin et al., 2014) dataset

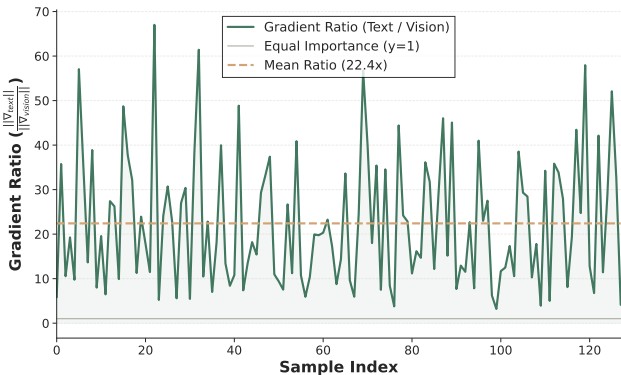

*Figure 4.* Analysis of gradient magnitude across 128 samples from the COCO (Lin et al., 2014) dataset. The text tokens exhibit significantly higher gradient norms compared to vision tokens, with an average ratio of 22.4.

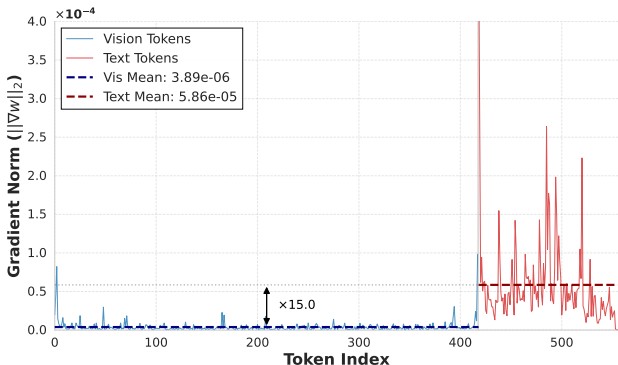

*Figure 5.* Detailed gradient analysis of a representative sample (Sample 88). The visualization highlights that the text-to-vision gradient ratio reaches approximately 15, confirming the dominance of textual information in the inference process.

reveals that the average gradient magnitude of text tokens exceeds that of visual tokens by a factor of **22.4**. This disparity is particularly pronounced in samples with extreme modality imbalance (e.g., containing only 5 text tokens against over 200 vision tokens), where the gradient ratio spikes drastically. This indicates that despite their scarcity, text tokens exert a disproportionately significant influence on the final generation results. Further inspection of a specific case study (Sample 88, detailed in Figure 5) corroborates this observation, showing an average text-to-vision gradient ratio of **15**. Such a gradient gap across diverse scenarios highlights a fundamental sensitivity imbalance between modalities.

This discrepancy implies that employing a uniform quantization strategy across all experts ignores modality-specific sensitivities. Treating the high-impact text tokens with the same granularity as redundant vision tokens potentially compromises performance, especially in cross-modal tasks where precise language generation is paramount.

### 3.1.2. EXPERTS' HETEROGENEITY

To understand the internal mechanism of cross-modal processing, we investigate the routing behaviors within the MoE layers. Our analysis focuses on the activation patterns and expert preferences across different modalities, revealing three key observations regarding expert affinity.

**Intrinsic Sparsity and Load Imbalance.** First, we analyze the inherent sparsity of expert activation. Regardless of the calibration dataset's composition, a persistent load imbalance is observed across the expert population. Notably, even when the calibration batch size is increased to 64 (approximately 30,000 tokens), certain experts receive zero input tokens. This phenomenon indicates that the non-uniform distribution of expert utilization is an intrinsic property of the pre-trained weights rather than an artifact of data sampling. Consequently, given that a select subset of experts dominates the model's output, applying a uniform metric to measure quantization error across all experts is suboptimal.

**Heterogeneity in Token Distribution.** Second, the analysis of routing paths reveals a nuanced functional division of labor, characterized by the coexistence of generalist and modality-specific experts. While a subset of experts acts as universal processors activated by both visual and textual inputs, others exhibit distinct routing preferences.

Specifically, as illustrated in Figure 6(e), we observe the spontaneous emergence of specialist clusters: some are dedicated to handling spatially redundant visual features, whereas others focus exclusively on the semantic density of textual information. This distribution confirms that the model allocates capacity both for cross-modal alignment and for modality-specific processing. The distinct activation frequencies and magnitudes across these clusters underscore the uneven contribution of different experts. Consequently, it is imperative to assign quantization error weights according to these modality-dependent activation characteristics, ensuring that the optimization strategy adapts to the specific functional role of each expert.

**Routing Bias and Decisive Experts.** Finally, the router outputs demonstrate significant bias, where a small fraction of experts plays a decisive role in the model's output. The routing probability distribution is highly skewed; for any given token, the router assigns high confidence scores to only a few experts (the top-$k$ selection), while the affinity for the remaining experts approaches zero. As indicated by the gradient magnitude analysis, these high-affinity experts dominate the inference outcome, whereas the influence of the non-selected experts is negligible. This implies that preserving the precision of these decisive experts is critical for maintaining model performance.

### 3.2. Modality-Expert-Aware Quantization

Building upon the analysis in Section 3.1, we have established that experts within MoE layers exhibit significant heterogeneity in both activation frequency and modality-specific sensitivity.Representative PTQ frameworks, such as

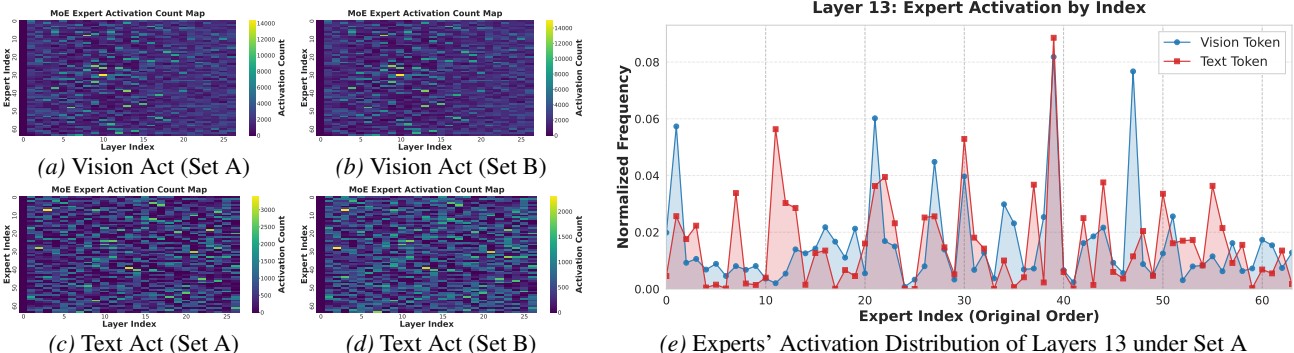

*Figure 6.* Visualization of expert affinity patterns. Subplots (a)-(d) illustrate the distinct activation distributions for vision and text tokens across different input ranges of the **COCO dataset** (Lin et al., 2014), highlighting how expert activation varies with different input samples while maintaining sparsity and modality-specific clustering. Subplot (e) visualizes the activation characteristics of the 13th layer in Kimi-VL-Instruct (Team et al., 2025) under Set A, highlighting the intrinsic load imbalance.

AWQ (Lin et al., 2024), typically determine optimal quantization parameters via a grid search aimed at minimizing generic reconstruction loss. However, such a search process treats all experts indiscriminately, failing to account for their varying importance during inference.

To address this, we propose a Modality-Expert-Aware Quantization method that incorporates expert importance into the error minimization objective.

**Quantifying Expert Importance.** First, we define an importance score $S_i$ for the $i$-th expert as a balanced measure of its contribution across modalities. Due to the inherent modality imbalance in MoE VLMs, where vision tokens significantly outnumber text tokens, a raw frequency count would disproportionately favor vision-dominant experts.

Therefore, we introduce a comprehensive normalization mechanism to better align the magnitude of contributions from different modalities. The importance score $S_i$ is formulated as a weighted summation:

$$S_i = \gamma \cdot N_i^{\text{text}} + \beta \cdot N_i^{\text{vis}}, \qquad (1)$$

where $N_i^{\text{text}}$ and $N_i^{\text{vis}}$ denote the number of text and visual tokens routed to the $i$-th expert, respectively. Let $T_{\text{text}}$ and $T_{\text{vis}}$ represent the total number of text and vision tokens in the calibration set, respectively. The coefficient $\beta = T_{\text{text}}/T_{\text{vis}}$ serves as a **quantity normalization factor**, scaling down the frequent visual activations to be comparable with textual counts. The coefficient $\gamma = \|\nabla_{\text{text}}\|/\|\nabla_{\text{vis}}\|$ acts as a **quality sensitivity factor**, reflecting the higher gradient impact of text tokens. This formulation ensures that the resulting score is driven by the semantic significance of the tokens rather than their raw frequency, protecting the decisive experts responsible for logical reasoning.

**Importance-Aware Optimization Objective.** As mentioned above, established PTQ strategies like AWQ (Lin et al., 2024) optimize quantization scales through grid-based search, treating the MoE layer as a monolithic entity. These methods apply a uniform optimization target across the

model. Specifically within the MoE module, this search process is equivalent to minimizing an unweighted summation of reconstruction errors across all experts:

$$\mathcal{L}_{\text{Standard}} = \sum_{i=1}^{M} \|\mathbf{W}_i \mathbf{X}_i - \hat{\mathbf{W}}_i \mathbf{X}_i\|_F^2, \qquad (2)$$

where $\mathbf{W}_i$ and $\hat{\mathbf{W}}_i$ denote the full-precision and quantized weights of the $i$-th expert, respectively, $\mathbf{X}_i$ represents the input tokens routed to that expert, and $M$ is the total number of experts. This formulation treats rarely used experts and critical experts indistinguishably, failing to account for their varying contributions to the model's final inference results.

In contrast, our proposed method redefines the optimization objective by introducing the importance score $S_i$ (defined in Eq. 1) as a weighting factor. The weighted quantization error is expressed as the following equation:

$$\mathcal{L}_{\text{Weighted}} = \sum_{i=1}^{M} S_i \cdot \|\mathbf{W}_i \mathbf{X}_i - \hat{\mathbf{W}}_i \mathbf{X}_i\|_F^2. \qquad (3)$$

By minimizing $\mathcal{L}_{\text{Weighted}}$ during the search for optimal quantization parameters, quantization noise is preferentially suppressed in experts that exhibit high activation frequencies and high modality sensitivities. This ensures that the quantization parameters are calibrated to preserve the functionality of the most influential experts, thereby accurately reflecting the true impact of quantization on the model's overall performance on downstream tasks.

### 3.3. Modality-Affinity-Aware Quantization

Some other representative post-training quantization frameworks, such as GPTQ (Frantar et al., 2022), typically rely on second-order information to determine the optimal quantization parameters. Specifically, they utilize the Hessian matrix $H$, approximations of which are computed via input calibration data $X$ as $H = 2XX^\top$. This formulation implicitly assumes that all input tokens contribute equally to the reconstruction error, treating the optimization landscape as uniform across the sequence dimension.

**Affinity and Modality Imbalance.** However, as analyzed in Section 3.1, this assumption of uniformity is not suitable for MoE VLMs. The input tokens exhibit significant variance in their interaction with experts: (1) Routing Diversity: Tokens possess varying levels of affinity with specific experts, determined by the router's output probabilities; (2) Modality Sensitivity: Text tokens, despite being fewer in number, carry higher gradient densities and information value compared to spatially redundant vision tokens.

Directly applying a uniform Hessian calculation to the MoE layers fails to account for these nuances, potentially causing the neglect of critical semantic information.

**Enhanced Hessian Matrix.** To address this, we propose an effective Modality-Affinity-Aware formulation for the Hessian matrix. We introduce a token-wise importance weighting mechanism. Let $X \in \mathbb{R}^{d \times N}$ denote the input tokens specifically routed to the current expert, where $N$ is the number of such tokens. We reconstruct the Hessian matrix $\tilde{H}$ by scaling the contribution of each token according to its modality-specific affinity:

$$\tilde{H} = (X \cdot \sqrt{\mathbf{C}})(X \cdot \sqrt{\mathbf{C}})^\top = X\mathbf{C}X^\top, \qquad (4)$$

where $\mathbf{C} \in \mathbb{R}^{N \times N}$ is a diagonal matrix representing the importance weight for each token. For the $j$-th token $x_j$, the corresponding diagonal element $c_j$ is defined as:

$$c_j = p_j \cdot \alpha_j, \quad \text{where } \alpha_j = \begin{cases} \gamma & x_j \text{ is text token,} \\ 1 & x_j \text{ is vision token.} \end{cases} \qquad (5)$$

Here, $p_j$ denotes the affinity between token $j$ and the target expert. The term $\gamma$ represents the gradient scaling factor, formally defined as the ratio of gradient magnitudes between textual and visual modalities: $\gamma = \|\nabla_{\text{text}}\| / \|\nabla_{\text{vis}}\|$.

By incorporating $\mathbf{C}$ into the Hessian computation, the quantization objective is dynamically re-weighted. Tokens with high router affinity and high modality sensitivity exert a larger influence on the structure of $\tilde{H}$. Consequently, when the Inverse Hessian is applied to update the weights, the algorithm prioritizes preserving the accuracy of these high-impact tokens, thereby maximizing the model's performance retention under low-bit settings.

## 4. Experiments

### 4.1. Experiment setting

We conduct comprehensive experiments to evaluate the efficacy of our proposed quantization method.

**Model and Benchmarks.** We choose to utilize Kimi-VL-Instruct (Team et al., 2025) and Qwen3-VL-30B-A3B-Instruct (Bai et al., 2025) for experiments. To ensure a robust assessment of multimodal capabilities, we employ a diverse set of widely recognized benchmarks, including MMMU (Yue et al., 2024) and MMBench (Liu et al., 2024a) for multi-discipline reasoning, AI2D (Kembhavi et al., 2016)

for diagram understanding, InfoVQA (Mathew et al., 2022) and TextVQA (Singh et al., 2019) for OCR-related visual question answering, as well as MME-RealWorld (Zhang et al., 2024b), RealWorldQA (Team, 2024), ScienceQA (Lu et al., 2022) and VizWiz-VQA (Gurari et al., 2019) to cover real-world and scientific scenarios.

**Baselines.** We compare our method against the full-precision version and several established quantization baselines to demonstrate its superiority: 1) BF16: The original model in Bfloat16 precision, serving as the performance upper bound; 2) RTN (Round-to-Nearest): A naive baseline that quantizes weights by rounding them to the nearest grid point; 3) Advanced PTQ Frameworks: We include GPTQ (Frantar et al., 2022) and AWQ (Lin et al., 2024), which are widely adopted as standard baselines for LLM compression, alongside MBQ (Li et al., 2025), the current SOTA method specifically tailored for VLM quantization.

**Implementation Details.** Our evaluation pipeline is built upon the open-source lmms-eval (Zhang et al., 2024a) framework, a standardized toolkit designed for the rigorous evaluation of Large Multimodal Models. For the inference backend, we employ SGLang (Zheng et al., 2024), a high-throughput serving engine optimized for MoE architectures to make inference efficient. During evaluation process, model outputs are generated and then compared against the standard ground-truth answers for each benchmark to compute the final accuracy scores.

### 4.2. Main Results

We conduct a comprehensive evaluation of our proposed method against state-of-the-art baselines on the Kimi-VL-Instruct (Team et al., 2025) and Qwen3-VL-30B-A3B-Instruct (Bai et al., 2025) models. For clarity in the following analysis, we denote the implementation of **Modality-Expert-Aware Quantization** based on AWQ (Lin et al., 2024) as VEQ-ME, while the version incorporating **Modality-Affinity-Aware Quantization** based on GPTQ is referred to as VEQ-MA. To fully assess the robustness of quantization, we perform experiments under both 4-bit (W4) and 3-bit (W3) weight quantization settings. The detailed comparison results across seven multimodal benchmarks are presented in Table 1.

**Performance under W4 Setting.** As shown in the upper section of Table 1, most methods maintain robust performance under the 4-bit quantization setting. Specifically, for the Kimi-VL-Instruct model, established baselines such as AWQ (Lin et al., 2024), MBQ (Li et al., 2025) and GPTQ (Frantar et al., 2022), along with our proposed VEQ, recover nearly 98% of the average BF16 accuracy. The relatively marginal performance gap among different quantization strategies suggests that 4-bit precision offers sufficient capacity to represent MoE weights without incurring catastrophic information loss.

*Table 1.* Main comparison results of Kimi-VL-Instruct and Qwen3-VL-30B-A3B-Instruct under 3-bit (W3) and 4-bit (W4) weight only quantization. We report the zero-shot accuracy (%) for all tasks. The best results for each bit-width are highlighted in **bold**. The improvement of VEQ-MA over the best baseline method is marked in parentheses. Abbreviations: InfoV: InfoVQA, TextV: TextVQA, RWQA: RealWorldQA, SciQA: ScienceQA, Viz: VizWiz, MMB: MMBench, MME-R: MME-RealWorld.

| Model | Bit | Method | Benchmarks | | | | | | | | | Avg. |
|---|---|---|---|---|---|---|---|---|---|---|---|---|
| | | | MMMU | AI2D | InfoV | TextV | RWQA | SciQA | Viz | MMB | MME-R | |
| Kimi-VL-Instruct | **BF16** | - | 51.11 | 83.65 | 83.38 | 85.93 | 66.41 | 92.10 | 69.00 | 82.73 | 58.22 | 74.73 |
| | W3 | RTN | 37.00 | 69.14 | 59.68 | 74.58 | 56.47 | 79.06 | 58.51 | 66.06 | 46.13 | 60.74 |
| | | AWQ | 41.78 | 70.05 | 61.49 | 74.04 | 58.17 | 81.84 | 61.37 | 71.13 | 50.44 | 63.37 |
| | | MBQ | 39.76 | 70.89 | 58.86 | 74.35 | 58.74 | 81.82 | 60.87 | 71.13 | 49.91 | 62.93 |
| | | GPTQ | 40.33 | 75.49 | 64.14 | 64.30 | 58.82 | 83.87 | 55.61 | 73.02 | 45.41 | 62.33 |
| | | VEQ-ME (ours) | **44.56** | 71.57 | 62.33 | 73.36 | **58.95** | 82.93 | 58.90 | 71.91 | **51.46** | 64.00 |
| | | VEQ-MA (ours) | 42.56 | **75.65** | **64.48** | **78.30** | 58.30 | **84.85** | **63.10** | **73.42** | 48.03 | **65.41** (+2.04) |
| | W4 | RTN | 48.44 | 82.38 | 80.16 | 84.60 | **66.80** | 90.83 | **69.17** | 80.67 | 52.48 | 72.84 |
| | | AWQ | 49.00 | 81.74 | 79.01 | 83.37 | 66.27 | 90.92 | 69.07 | 80.58 | 53.70 | 72.63 |
| | | MBQ | 48.89 | 81.70 | 78.49 | 83.18 | 64.58 | 90.29 | 69.03 | 80.93 | 55.08 | 72.46 |
| | | GPTQ | 50.00 | 82.32 | 80.04 | 84.63 | 64.84 | 91.39 | 68.86 | 81.44 | 53.14 | 72.96 |
| | | VEQ-ME (ours) | 49.22 | 81.41 | 79.75 | 83.69 | 66.36 | 91.06 | 69.17 | 80.84 | 54.20 | 72.86 |
| | | VEQ-MA (ours) | **50.30** | **82.55** | **80.37** | **84.71** | 66.41 | **91.56** | 68.91 | **81.67** | **55.61** | **73.57** (+0.61) |
| Qwen3-VL-30B-A3B-Instruct | **BF16** | - | 73.67 | 86.27 | 81.43 | 81.31 | 65.49 | 93.52 | 83.27 | 85.91 | 59.92 | 78.98 |
| | W3 | RTN | 57.33 | 77.10 | 44.96 | 68.64 | 43.01 | 89.08 | 64.07 | 78.60 | **45.73** | 63.17 |
| | | AWQ | 58.89 | 73.61 | 44.62 | 67.93 | 45.36 | 88.40 | 61.75 | 77.06 | 42.38 | 62.22 |
| | | MBQ | 50.56 | 71.44 | 40.15 | 64.23 | 53.82 | 87.31 | 59.48 | 76.46 | 45.27 | 60.97 |
| | | GPTQ | 64.89 | 74.11 | **47.27** | 71.82 | 54.25 | 88.23 | 63.39 | 69.39 | 44.08 | 64.16 |
| | | VEQ-ME (ours) | 60.56 | 73.38 | 44.95 | 69.91 | 53.73 | 88.40 | 62.47 | 77.75 | 44.24 | 63.93 |
| | | VEQ-MA (ours) | **65.89** | **79.15** | 47.14 | **72.74** | **54.77** | **89.55** | **69.46** | **82.30** | 43.25 | **67.14** (+3.09) |
| | W4 | RTN | 63.11 | **83.33** | 62.62 | 79.77 | 45.23 | 91.51 | 71.08 | 85.13 | 57.31 | 70.89 |
| | | AWQ | 65.20 | 81.22 | 58.32 | 77.40 | 63.14 | 91.06 | 69.73 | 85.22 | 57.31 | 72.07 |
| | | MBQ | 69.67 | 81.41 | 59.36 | 78.99 | 64.31 | 91.51 | 70.78 | 83.68 | 56.10 | 72.87 |
| | | GPTQ | **72.67** | 82.93 | 62.54 | 80.18 | 53.46 | **93.23** | 71.32 | 84.53 | 57.96 | 73.20 |
| | | VEQ-ME (ours) | 68.78 | 82.47 | 59.53 | 78.57 | **65.32** | 91.61 | 70.97 | 83.56 | 56.79 | 73.07 |
| | | VEQ-MA (ours) | 71.56 | 82.95 | **62.86** | **81.10** | 61.70 | 92.64 | **72.42** | **85.88** | **58.12** | **74.36** (+1.16) |

**Performance under W3 Setting.** The distinction between methods becomes significantly more pronounced in the aggressive 3-bit setting. As shown in the lower section of Table 1, traditional baselines such as RTN and AWQ (Lin et al., 2024) suffer from severe degradation, particularly on reasoning-intensive tasks like MMMU (Yue et al., 2024) and fine-grained visual tasks like InfoVQA (Mathew et al., 2022). This suggests that the uniform quantization assumptions fail when the bit-width is extremely limited. It also proves that ignoring the heterogeneity can lead to errors under extreme compression. In contrast, VEQ demonstrates exceptional robustness. By explicitly modeling expert importance and modality affinity, VEQ significantly outperforms the baselines. For instance, on TextVQA (Singh et al., 2019), VEQ achieves a gain of **21.4%** compared to the original quantization method on Kimi-VL-Instruct. These results validate that protecting the decisive experts and differentiating modality sensitivities are critical in low-bit regimes.

### 4.3. Ablation Studies

To verify the effectiveness and robustness of our proposed method, we conduct a two-fold ablation study. First, we evaluate the contribution of each component to the overall performance on downstream tasks. Second, we perform a hyperparameter sensitivity analysis on a randomly extracted validation set to justify our parameter selection. In addition, we report all ablation results under the same quantization configuration to ensure a fair comparison. We further keep the calibration data and evaluation protocol fixed across settings to isolate the effect of each component.

**Component Effectiveness on Downstream Tasks.** We focus on the two core formulations: the Expert Importance Score in VEQ-ME and the Modality-Affinity-Aware Hessian in VEQ-MA. We set the hyperparameters to their default optimal values and measure the performance drop when specific components are disabled.

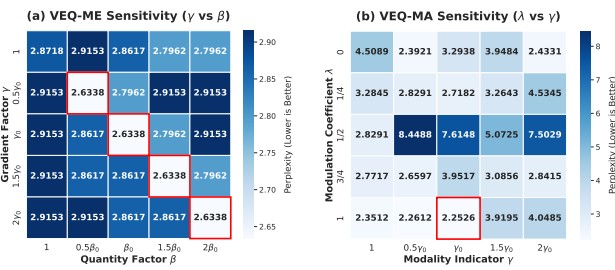

*(a) VEQ-ME Sensitivity.*     *(b) VEQ-MA Sensitivity.*

*Figure 7.* Visual analysis of parameter sensitivity regarding validation PPL. (a) VEQ-ME: It confirms the scale invariance of our method, where maintaining the relative ratio ensures consistent minimization of quantization error. (b) VEQ-MA: The results shows reducing $\lambda$ generally results in an increase in PPL, validating that the router confidence is important for accurate quantization.

*Table 2.* Ablation study of Modality-Expert Importance (VEQ-ME) on downstream tasks. $\gamma$: Gradient factor; $\beta$: Quantity factor.

| Config | MMMU | InfoVQA | ScienceQA | Avg. |
|---|---|---|---|---|
| w/o $\gamma$ ($\gamma = 1$) | 43.44 | 60.39 | 82.24 | 62.02 |
| w/o $\beta$ ($\beta = 1$) | 43.89 | 60.34 | 82.67 | 62.30 |
| **VEQ-ME (Full)** | **44.56** | **62.33** | **82.93** | **63.27** |

*1) Impact of Modality-Expert Importance (VEQ-ME).* The importance score is formulated as $S_i = \gamma N_i^{text} + \beta N_i^{vision}$. We investigate the necessity of the gradient scaling factor $\gamma$ and the quantity normalization factor $\beta$:

- **w/o $\gamma$ ($\gamma = 1$):** Removing the gradient scale treats visual and textual importance purely based on token quantity. As shown in Table 2, this leads to a noticeable drop in text-heavy reasoning tasks, confirming that text tokens require higher sensitivity weights.

- **w/o $\beta$ ($\beta = 1$):** Ignoring the quantity gap allows the vast number of vision tokens to dominate the score. The results indicate that this variation degrades performance, as the router becomes biased toward spatially redundant visual features.

*2) Impact of Modality-Affinity Awareness (VEQ-MA).* For affinity-aware quantization, the token weight is defined as $c_j = p_j \cdot \alpha_j$. We examine the roles of affinity $p_j$ and the modality indicator $\alpha_j$:

- **w/o $p$ ($p = 1$):** Ignoring router affinity logits treats all tokens routed to an expert as equally important. Table 3 shows that this leads to suboptimal results, proving that tokens with higher routing confidence are more representative.

- **w/o $\alpha$ ($\alpha = 1$):** Removing modality-specific re-weighting during Hessian calibration causes a performance decline, confirming that distinguishing between information-dense text tokens and redundant vision tokens is essential for accurate error minimization.

*Table 3.* Ablation study of Affinity-Aware Hessian (VEQ-MA) on downstream tasks. $p$: Router confidence; $\alpha$: Modality indicator.

| Config | MMMU | InfoVQA | ScienceQA | Avg. |
|---|---|---|---|---|
| w/o $p$ ($p = 1$) | 42.33 | 64.30 | 83.35 | 63.63 |
| w/o $\alpha$ ($\alpha = 1$) | 41.33 | 63.72 | 82.34 | 62.46 |
| **VEQ-MA (Full)** | **42.56** | **64.48** | **84.85** | **63.96** |

**Parameter Sensitivity Analysis.** To further validate the robustness of our method, we analyze the average Perplexity (PPL) on 64 samples under varying parameter configurations. These samples are randomly extracted from MMMU (Yue et al., 2024) validation dataset. We perform a grid search for the variable pairs in VEQ-ME ($\gamma, \beta$) and VEQ-MA ($\lambda, \gamma$). It is worth noting that while the formulation of VEQ-MA is theoretically governed by the affinity $p$ and sensitivity ratio $\alpha$, directly tuning $p$ and $\alpha$ lacks intuitive interpretability. Consequently, we choose to adopt ($\lambda, \gamma$) as the variable pair for this ablation study.

*1) Sensitivity of VEQ-ME ($\gamma$ vs. $\beta$).* In our formulation, $\gamma$ represents the sensitivity of text tokens relative to visual tokens, while $\beta$ accounts for the quantity ratio between the two modalities. These hyperparameters determine the expert importance, which guides the search for the optimal quantization parameters. As illustrated in Figure 7, we observe that proportionally scaling $\gamma$ and $\beta$ yields consistent PPL values. This implies that the search for quantization parameters depends on the *relative* ratio of expert importance.

*2) Sensitivity of VEQ-MA ($\lambda$ vs. $\gamma$).* We analyze the Hessian weighting by varying the modality sensitivity ratio $\alpha$ and the affinity strength $p$. Specifically, we treat the variation of $p$ using a modulation coefficient $\lambda$, which controls the intensity of router confidence. We define the effective affinity as a weighted interpolation: $(1 - \lambda) \cdot \mathbf{1} + \lambda \cdot \mathbf{p}$. Under this formulation, $\lambda = 1$ represents the raw router confidence, while $\lambda = 0$ indicates uniform affinity. As shown in Figure 7, the model achieves optimal stability (lowest PPL of 2.2526) at the full configuration ($\lambda = 1, \gamma = \gamma_0$).

## 5. Conclusion

In this work, we presented **V**isual **E**xpert **Q**uantization (**VEQ**), a specialized post-training quantization framework designed to address the unique challenges of compressing Mixture-of-Experts Vision-Language Models (MoE VLMs). By transcending the limitations of treating MoE FFNs as monolithic dense structures, VEQ effectively addresses both the inherent sparsity of expert activations and the statistical heterogeneity between visual and textual modalities. Across a diverse set of multimodal benchmarks, VEQ consistently outperforms established baselines. By aligning quantization strategies with the structural and modal properties of MoE VLMs, VEQ establishes a new state-of-the-art, paving the way for the efficient deployment of large-scale multimodal agents in resource-constrained environments.

## Impact Statement

This paper presents work whose goal is to advance the field of Machine Learning. There are many potential societal consequences of our work, none of which we feel must be specifically highlighted here.

## Acknowledgments

This work is supported by the National Natural Science Foundation of China (62501386, 625B2116), CCF-Tencent Rhino-Bird Open Research Fund, and CAAI-Tencent Rhino-Bird Open Research Fund. This work is also sponsored by Al Hundred Schools Program and is carried out using the Ascend AI technology stack.

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
