# Supplementary Material for
# VEQ: Modality-Adaptive Quantization for MoE Vision-Language Models

## 1. Preliminaries

### 1.1. Model Quantization

Quantization aims to map high-precision floating-point values (e.g., FP16 or BF16) to lower-precision integers (e.g., INT4 or INT8) to reduce memory footprint and accelerate inference. Typical weight-only quantization methods apply asymmetric uniform quantization for model compression. Given a high-precision weight tensor $\mathbf{W}$, the quantization parameters, namely the scaling factor $s$ and the zero-point $z$, are defined based on the dynamic range of the weights:

$$s = \frac{\max(\mathbf{W}) - \min(\mathbf{W})}{q_{\max} - q_{\min}},$$
$$z = \text{round}\left(q_{\min} - \frac{\min(\mathbf{W})}{s}\right). \tag{1}$$

where $[q_{\min}, q_{\max}]$ represents the valid range of the quantized integer (e.g., $[-8, 7]$ for 4-bit signed integers). Using these parameters, the floating-point weights $\mathbf{W}$ are mapped to integer representations $\mathbf{W}_{\text{int}}$:

$$\mathbf{W}_{\text{int}} = \text{clamp}\left(\left\lfloor \frac{\mathbf{W}}{s} \right\rceil + z, q_{\min}, q_{\max}\right), \tag{2}$$

where $\lfloor \cdot \rceil$ denotes the nearest rounding operation. The dequantized approximation $\hat{\mathbf{W}}$ is then reconstructed as:

$$\hat{\mathbf{W}} = s \cdot (\mathbf{W}_{\text{int}} - z). \tag{3}$$

For state-of-the-art post-training quantization methods (e.g., GPTQ, AWQ), the goal is to find the optimal quantization parameters and quantized weights that minimize the reconstruction error of layer outputs. This is typically formulated as follows:

$$\min_{\hat{\mathbf{W}}} \|\mathbf{W}\mathbf{X} - \hat{\mathbf{W}}\mathbf{X}\|_F^2, \tag{4}$$

where $\mathbf{W}$ represents the original weights, $\mathbf{X}$ represents the input calibration data (activations), and $\|\cdot\|_F$ denotes the Frobenius norm. While this objective treats all parameters uniformly, our proposed method revisits this assumption in the context of MoE architectures.

### 1.2. MoE Architecture and Gating Mechanism

The Mixture-of-Experts (MoE) architecture replaces the standard Feed-Forward Network (FFN) in Transformers with a sparse layer containing multiple independent networks, referred to as experts. Let $\{E_i\}_{i=1}^N$ denote a set of $N$ experts. For an input token representation $\mathbf{x} \in \mathbb{R}^d$, the selection of experts is governed by a gating network (or router). The gating score is typically computed via a linear transformation followed by a Softmax function:

$$S(\mathbf{x}) = \text{Softmax}(\mathbf{x}\mathbf{W}_g), \tag{5}$$

---

[1]Anonymous Institution, Anonymous City, Anonymous Region, Anonymous Country. Correspondence to: Anonymous Author <anon.email@domain.com>.

where $\mathbf{W}_g \in \mathbb{R}^{d \times N}$ is the trainable weight matrix of the router. To ensure sparsity and computational efficiency, modern MoE LLMs (e.g., Kimi-VL (Team et al., 2025), Qwen3-VL (Bai et al., 2025)) utilize a Top-$k$ routing strategy. Instead of computing the output for all experts, the router selects a subset of indices $\mathcal{T}$ corresponding to the experts with the highest gating scores:

$$\mathcal{T} = \text{Top-}k(S(\mathbf{x})). \tag{6}$$

Consequently, the final output of the MoE layer $\mathbf{y}$ is calculated as the weighted sum of the selected experts' outputs:

$$\mathbf{y} = \sum_{i \in \mathcal{T}} S(\mathbf{x})_i \cdot E_i(\mathbf{x}). \tag{7}$$

This sparse activation mechanism implies that different experts have varying degrees of impact on the model's output depending on the input distribution, a property that our method VEQ exploits for optimized quantization.

## 2. Mathematical Justification

In this section, we provide a mathematical justification that our two reweighting strategies (i) Modality-Expert-Aware Quantization and (ii) Modality-Affinity-Aware Quantization, indeed reduce a well-defined *quantization surrogate loss*. Importantly, our claims are rigorous with respect to these surrogate objectives. Connecting them to the true downstream task loss additionally relies on standard smoothness/second-order approximations, which we state explicitly.

**Notation (Top-k Weighted MoE).** Consider a Top-$k$ MoE layer and $M$ experts. For token $t$, the router selects a top-k index set $\mathcal{K}_t \subset \{1, \ldots, M\}$ and assigns mixture weights $a_{t,i} \geq 0$ such that $\sum_{i \in \mathcal{K}_t} a_{t,i} = 1$ (and $a_{t,i} = 0$ for $i \notin \mathcal{K}_t$). Let $x_t \in \mathbb{R}^d$ be the expert input of token $t$ and $W_i \in \mathbb{R}^{d_o \times d}$ be the full-precision weight of expert $i$. Under weight-only quantization, the quantized weight is $\hat{W}_i = W_i + \Delta W_i$. The MoE output for token $t$ is

$$y_t = \sum_{i \in \mathcal{K}_t} a_{t,i} W_i x_t, \qquad \hat{y}_t = \sum_{i \in \mathcal{K}_t} a_{t,i} \hat{W}_i x_t, \tag{8}$$

and thus the output perturbation induced by quantization is

$$\delta y_t := \hat{y}_t - y_t = \sum_{i \in \mathcal{K}_t} a_{t,i} \Delta W_i x_t. \tag{9}$$

### 2.0.1. EXPERT-IMPORTANCE WEIGHTING MINIMIZES A PRINCIPLED UPPER BOUND

**A decomposition upper bound for Top-k MoE.** Since $\| \cdot \|_2^2$ is convex, Jensen's inequality yields, for any vectors $\{v_i\}$, $\| \sum_i a_i v_i \|_2^2 \leq \sum_i a_i \| v_i \|_2^2$ when $a_i \geq 0$ and $\sum_i a_i = 1$. Applying this to Eq. (9) with $v_i = \Delta W_i x_t$, we obtain for each token $t$,

$$\|\delta y_t\|_2^2 = \Big\| \sum_{i \in \mathcal{K}_t} a_{t,i} \Delta W_i x_t \Big\|_2^2 \leq \sum_{i \in \mathcal{K}_t} a_{t,i} \|\Delta W_i x_t\|_2^2. \tag{10}$$

Summing over all calibration tokens gives

$$\sum_t \|\delta y_t\|_2^2 \leq \sum_t \sum_{i \in \mathcal{K}_t} a_{t,i} \|\Delta W_i x_t\|_2^2 = \sum_{i=1}^{M} \sum_t a_{t,i} \|\Delta W_i x_t\|_2^2. \tag{11}$$

Eq. (11) shows that the true MoE output reconstruction error is upper bounded by a sum of per-expert errors weighted by router mixture coefficients.

**Soft activation counts and expert importance.** For Top-k routing, we define *soft* modality-specific activation counts for expert $i$ as

$$N_i^m := \sum_{t \in m} a_{t,i}, \qquad m \in \{\text{text, vis}\}, \tag{12}$$

and the expert importance score as

$$S_i := \gamma N_i^{\text{text}} + \beta N_i^{\text{vis}}, \tag{13}$$

where $\beta$ compensates the extreme imbalance in token quantities and $\gamma$ reflects the larger modality sensitivity of text tokens.

**Weighted surrogate objective.** Motivated by Eq. (11), we define the following expert-importance-weighted surrogate loss:

$$\mathcal{L}_{\text{Weighted}} := \sum_{i=1}^{M} S_i \sum_t a_{t,i} \|\Delta W_i x_t\|_2^2. \tag{14}$$

Compared with uniform PTQ objectives, Eq. (14) explicitly prioritizes experts that are both frequently used (in the soft-count sense) and modality-sensitive.

**Theorem 1 (Optimality w.r.t. the weighted surrogate).** Let $\mathcal{Q}$ be the feasible set of weight-only quantized weights (e.g., determined by bitwidth and scale/zero-point parameterization). Define the quantized experts by

$$\{\hat{W}_i^\star\}_{i=1}^{M} = \arg \min_{\{\hat{W}_i\} \in \mathcal{Q}} \mathcal{L}_{\text{Weighted}}(\{\hat{W}_i\}). \tag{15}$$

Then for any other feasible quantized weights $\{\tilde{W}_i\} \in \mathcal{Q}$,

$$\mathcal{L}_{\text{Weighted}}(\{\hat{W}_i^\star\}) \leq \mathcal{L}_{\text{Weighted}}(\{\tilde{W}_i\}). \tag{16}$$

**Proof.** Eq. (16) follows directly from the definition of the minimizer of $\mathcal{L}_{\text{Weighted}}$ over the feasible set $\mathcal{Q}$. $\square$

**Discussion (relation to downstream task loss).** Let $\mathcal{J}$ denote the downstream task loss and assume local second-order smoothness w.r.t. the MoE output, so that for small perturbations,

$$\Delta \mathcal{J} \approx \sum_t \frac{1}{2} \delta y_t^\top G_t \, \delta y_t, \tag{17}$$

where $G_t = \nabla_{y_t}^2 \mathcal{J}$. If the average curvature for text tokens is larger than that for vision tokens (captured by $\gamma$), then minimizing Eq. (14) provides a closer proxy to the expected $\Delta \mathcal{J}$ than a uniform objective, thereby reducing performance degradation in practice.

### 2.0.2. MODALITY–AFFINITY-AWARE HESSIAN IS EXACTLY A TOKEN-WEIGHTED QUADRATIC FORM

**Token-weighted reconstruction error and its quadratic form.** Consider a linear mapping $y = Wx$ (this applies per expert or per linear block) under weight-only quantization $\hat{W} = W + \Delta W$. Given calibration inputs $X = [x_1, \ldots, x_N] \in \mathbb{R}^{d \times N}$ and nonnegative token weights $c_j \geq 0$, define the token-weighted reconstruction error:

$$\mathcal{L}_C(\Delta W) := \sum_{j=1}^{N} c_j \|\Delta W x_j\|_2^2. \tag{18}$$

**Theorem 2 (Equivalence to a weighted Hessian).** Let $C = \text{diag}(c_1, \ldots, c_N) \succeq 0$ and define

$$\tilde{H} := XCX^\top. \tag{19}$$

Then

$$\mathcal{L}_C(\Delta W) = \text{tr}\left(\Delta W \, \tilde{H} \, \Delta W^\top\right). \tag{20}$$

**Proof.**

$$\sum_{j=1}^{N} c_j \|\Delta W x_j\|_2^2 = \sum_{j=1}^{N} c_j \, x_j^\top \Delta W^\top \Delta W x_j = \text{tr}\left(\Delta W^\top \Delta W \sum_{j=1}^{N} c_j x_j x_j^\top\right)$$
$$= \text{tr}\left(\Delta W^\top \Delta W \, XCX^\top\right) = \text{tr}\left(\Delta W \, (XCX^\top) \, \Delta W^\top\right), \tag{21}$$

which yields Eq. (20).

**Router affinity and modality sensitivity as token weights.** In MoE-based VLMs, tokens contribute unequally due to (i) router affinity and (ii) modality sensitivity. We set

$$c_j = p_j \alpha_j, \qquad \alpha_j = \begin{cases} \gamma, & x_j \text{ is a text token,} \\ 1, & x_j \text{ is a vision token,} \end{cases} \tag{22}$$

where $p_j$ is the router confidence/affinity (for Top-k, this can be instantiated as the selected mixture coefficient for the corresponding expert, e.g., $p_{t,i} = a_{t,i}$), and $\gamma$ scales the relative sensitivity of text tokens.

For Top-6 weighted routing, a more fine-grained expert-specific form is

$$c_{t,i} := a_{t,i} \alpha_t, \qquad \tilde{H}_i := \sum_t c_{t,i} x_t x_t^\top = X \operatorname{diag}(c_{\cdot,i}) X^\top, \tag{23}$$

so that each expert's curvature estimate is dominated by tokens that (a) actually use the expert with high mixture weight and (b) are modality-sensitive.

**Corollary 1.** If a second-order PTQ method (e.g., GPTQ-style) approximately minimizes the quadratic form induced by $\tilde{H}$ (or $\tilde{H}_i$ per expert) over the same feasible set $\mathcal{Q}$, then the resulting quantized weights achieve a lower token-weighted reconstruction surrogate $\mathcal{L}_C$ than using an unweighted Hessian (i.e., $C = I$) under the same optimization accuracy, because $\tilde{H}$ corresponds exactly to the objective in Eq. (18)–(20).

**Discussion.** Under the same smoothness assumption as Eq. (17), the downstream loss increase is governed by the curvature-weighted energy of $\delta y_t$. Choosing $c_j$ to approximate each token's expected contribution to $\Delta\mathcal{J}$ (via router affinity and modality-dependent sensitivity) yields a quadratic surrogate whose minimization is more aligned with minimizing the expected task loss increase, which empirically improves low-bit performance retention.

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

## Example 1. Kimi-VL-Instruct Quantized Results

**Question:** A simply supported beam is subjected to a linearly varying distributed load $q(x) = \frac{x}{L}q_0$ with maximum intensity $q_0$ at B. The beam has a length $L = 4$ m and rectangular cross section with a width of 200 mm and height of 300 mm. Determine the maximum permissible value for $q_0$ if the allowable normal stresses are 120 MPa.

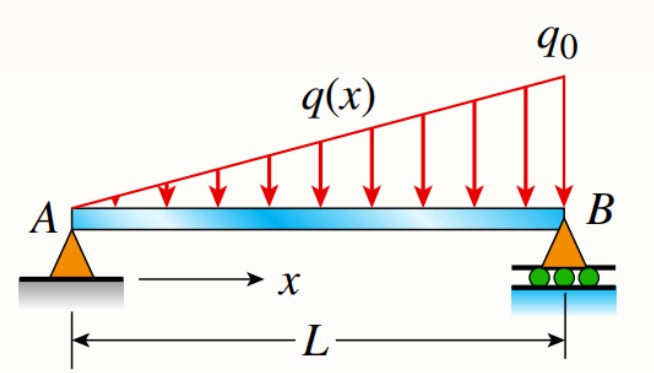

**Options:** [A: $q_0 = 250.7403\frac{kN}{m}$, B: $q_0 = 350.7403\frac{kN}{m}$, C: $q_0 = 450.7403\frac{kN}{m}$]

---

**GPTQ Response:** The correct answer is B. $q_0 = 350.7403\frac{kN}{m}$. This is because the maximum permissible value for the maximum intensity, $q_0$, can be calculated using the formula:
$q_0 = 120$ MPa $* 4$ m$/200$ mm $= 350.7403$ kN/m.
**\*\* Error Type: Process Error.** Although the final choice is correct, the calculation process is mathematically incorrect ($120 \times 4/200 \neq 350.7$).

---

**AWQ Response:** To determine the maximum permissible value for the maximum intensity $q_0$... [Content Omitted] ... The correct answer is to calculate the maximum permissible load based on the allowable stress and ensuring it does not exceed this maximum stress. The correct answer is to calculate...
**\*\* Error Type: Repetition Loop / Generation Collapse.**

---

**MBQ Response:** C
**\*\* Error Type: Incorrect Prediction.**

---

**Ours Response:** B. $q_0 = 350.7403\frac{kN}{m}$

### Example 2. Qwen3-VL-30B-A3B-Instruct Quantized Results

**Question:** Think about the magnetic force between the magnets in each pair. Which of the following statements is true?

*Hint: The images below show two pairs of magnets. The magnets in different pairs do not affect each other. All the magnets shown are made of the same material, but some of them are different shapes.*

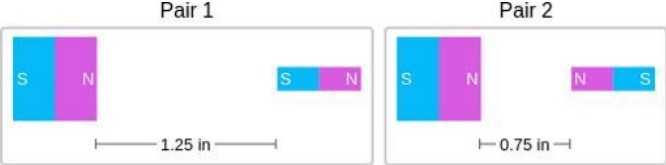

**Options:**

- A: The magnitude of the magnetic force is smaller in Pair 2.
- B: The magnitude of the magnetic force is smaller in Pair 1.
- C: The magnitude of the magnetic force is the same in both pairs.

-----------------------------------------------------------------------------------

**GPTQ Response:** A. The magnitude of the magnetic force is smaller in Pair 2.
**\*\* Error Type: Incorrect Prediction.**

-----------------------------------------------------------------------------------

**AWQ Response:** A.
**\*\* Error Type: Incorrect Prediction.**

-----------------------------------------------------------------------------------

**MBQ Response:** A. The magnitude of the magnetic force is smaller in Pair 2.
**\*\* Error Type: Incorrect Prediction.**

-----------------------------------------------------------------------------------

**Ours Response:** B