# OpenReview forum: "VEQ: Modality-Adaptive Quantization for MoE Vision-Language Models"
_ICML.cc/2026/Conference — ICML 2026 regular_

### Official Review · Reviewer_1Jse · 2026-03-11

**Soundness:** 3
**Presentation:** 4
**Significance:** 4
**Originality:** 4
**Overall Recommendation:** 5
**Confidence:** 2

**Summary:**

This work addresses the dual heterogeneity problem in post-training quantization of MoE-based vision–language models, including cross-modality differences and imbalanced expert contributions, by proposing a Visual Expert Quantization (VEQ) framework for models such as Kimi-VL and Qwen3-VL. VEQ introduces two components: VEQ-ME, which estimates expert importance using routing frequency and gradient sensitivity to weight the reconstruction loss, and VEQ-MA, which enhances the Hessian matrix with modality-aware router affinity to guide calibration. Under the W3A16 setting, VEQ consistently outperforms existing quantization methods such as AWQ, GPTQ, and MBQ, achieving average accuracy improvements of 2.04% on Kimi-VL and 3.09% on Qwen3-VL.

**Compliance With Llm Reviewing Policy:**

Affirmed.

**Final Justification:**

The problem has been basically solved.

**Key Questions For Authors:**

please see weakness

**Limitations:**

yes

**Strengths And Weaknesses:**

Strength:
1.  The authors provide a compelling analysis of the internal mechanics of cross-modal processing in MoE layers. They empirically prove that text tokens possess significantly higher gradient norms (an average ratio of 22.4x) compared to spatially redundant vision tokens. Furthermore, they highlight the intrinsic sparsity and routing bias where a small fraction of "decisive experts" dominates the model's output. This dual-heterogeneity perspective is highly original for VLM quantization
2. As MoE-based vision–language models such as Kimi-VL and the Qwen series become increasingly mainstream, VEQ enables W3 quantization to reduce memory usage to approximately one quarter while maintaining high accuracy, making it highly beneficial for edge deployment.

Weakness:
1. Only two models are evaluated, without covering a broader range of MoE-VLMs  or different model scales.
1. The empirical results show that the advantages of VEQ are mainly evident under extreme compression settings. In the W4 experiments, the performance differences between VEQ and existing methods such as AWQ, GPTQ, and MBQ are relatively small. Since 4-bit precision already provides sufficient representational capacity without severe information loss, the practical necessity of the proposed framework for standard W4 deployments appears somewhat limited.

---

> ### Author Rebuttal · Authors · 2026-03-31
>
> `Q1-1:`Only two models are evaluated, without covering a broader range of MoE-VLMs or different model scales.
>
> `A1-1:`We sincerely thank you for this constructive feedback regarding the scope of our evaluation. We fully agree that validating a proposed method across a broader spectrum of architectures and model scales is the ideal standard for comprehensive empirical research. However, we would like to respectfully clarify the practical and structural constraints that shaped our current experimental setup.
>
> First, the landscape of open-source Vision-Language Models that natively utilize the Mixture-of-Experts (MoE) architecture is still in its nascent stages. Consequently, the absolute number of publicly available and mature MoE-VLMs remains relatively small compared to the abundance of traditional dense models or text-only MoE LLMs. Second, the MoE architecture is predominantly employed precisely to scale models to massive parameter counts. Evaluating these ultra-large-scale MoE-VLMs demands an exceptionally high volume of computational resources and GPU memory. As an academic research group, scaling these experiments to cover a massive grid of ultra-large models currently exceeds our available hardware infrastructure.
>
> Given these objective constraints, we strategically selected two highly representative models for our evaluation. These models strictly encompass the state-of-the-art routing mechanisms, modality integration strategies, and structural bottlenecks prevalent in the current MoE-VLM paradigm, thereby serving as rigorous and robust testbeds to validate the core algorithmic contributions of our VEQ framework. We deeply appreciate the reviewer’s understanding of these computational limitations. In the revised manuscript, we will explicitly acknowledge this scope constraint in the Limitations section, highlighting the evaluation of future, diverse MoE-VLM families as an important direction for our subsequent work once more accessible models and compute resources become available.
>
> `Q1-2:`The empirical results show that the advantages of VEQ are mainly evident under extreme compression settings. In the W4 experiments, the performance differences between VEQ and existing methods such as AWQ, GPTQ, and MBQ are relatively small. Since 4-bit precision already provides sufficient representational capacity without severe information loss, the practical necessity of the proposed framework for standard W4 deployments appears somewhat limited.
>
> `A1-2:`
> Thank you for this perceptive observation. This phenomenon fundamentally aligns with the current developmental trajectory of the model compression field. Mainstream Post-Training Quantization (PTQ) methods have already achieved near-lossless performance at the 4-bit level for large-scale models. Consequently, the theoretical headroom for further algorithmic improvement within the W4 regime is inherently saturated.
>
> Therefore, the primary motivation and core contribution of our work intentionally target the extreme low-bit frontier. Exploring and stabilizing sub-4-bit compression is of critical practical value for the following reasons:
>
> 1. Mitigating the Memory Wall: Pushing the quantization boundary beyond 4-bit further drastically reduces the memory footprint. This is currently the most severe bottleneck for deploying massive, memory-bound MoE-based Vision-Language Models on consumer-grade GPUs or resource-constrained edge devices.
>
> 2. Future Hardware Co-design: While 4-bit represents the current optimal point for widespread hardware support, next-generation AI accelerators and emerging hardware architectures are actively exploring native support for sub-4-bit and mixed-precision data types. By successfully bridging the performance gap at these extreme low bit-widths today, our algorithmic framework paves the way for future hardware deployments to achieve multiplied inference efficiency and significantly reduced energy consumption.
>
> We deeply appreciate the reviewer highlighting this point. To provide a more comprehensive perspective, we will explicitly include this context as a discussion on limitations and future outlooks in the revised manuscript, clarifying that our method is explicitly designed to unlock the sub-4-bit frontier rather than incrementally optimizing the already saturated 4-bit space.

---

> > ### Author Rebuttal · Reviewer_1Jse · 2026-04-01
> >
> > The problem has been basically solved.

---

> > > ### Author Response · Authors · 2026-04-08
> > >
> > > Dear reviewer 1Jse,
> > >
> > > Thank you for your response. We are delighted to see that our answers were able to address your concerns.
> > >
> > > Best,
> > >
> > > Authors

---

### Official Review · Reviewer_ZUTV · 2026-03-12

**Soundness:** 2
**Presentation:** 2
**Significance:** 3
**Originality:** 3
**Overall Recommendation:** 4
**Confidence:** 4

**Summary:**

The paper proposes VEQ, a new quantization method for MoE VLM. VEQ takes into account the fact that some experts are used more frequently than others and that some tokens are more important than others during quantization. Numerous experiments on several benchmarks and MoE VLM show superior performance compared to the baselines considered.

**Compliance With Llm Reviewing Policy:**

Affirmed.

**Final Justification:**

The rebuttal addressed some of my main concerns.

**Key Questions For Authors:**

- Could you provide more interpretation on the new Hessian proposed in equation (4)? What loss function does this correspond to? Why does this loss function make sense?
- Could you add results for SpinQuant? And potentially your method applied to SpinQuant?
- Why not propose a framework that addresses both problems at the same time? For example, could the loss in (3) be changed by adding a C_i?
- Could you add error bars (from 3 or 5 seeds)?

**Limitations:**

yes

**Strengths And Weaknesses:**

**Soundness:**

The scientific approach is rigorous, and VEQ is motivated by numerous experiments and detailed observations. VEQ takes into account both the importance of vision/text tokens and the importance of experts in improving quantization quality.

However, there are a few weaknesses:
- The motivation for gamma in (1) is not very clear to me.
- H = 2XX^T is not an approximation (line 270). The Hessian has exactly this expression in the case of layer-wise reconstruction error.
- GPTQ is very competitive in Table 1. However, as mentioned in the paper, many methods improve on GPTQ by using rotation matrices. It would be more interesting to compare VEQ with methods such as SpinQuant. Furthermore, it seems that VEQ could also be applied to SpinQuant.
- The proposed new Hessian lacks interpretation (for example, what loss function does it correspond to?).
- The experiments lack uncertainty bars.
- The proposed method addresses either the fact that different experts may be more or less important (VEQ-ME) or the fact that some tokens are more important than others (VEQ-MA), but not both at the same time.

**Presentation:**

The presentation is good overall. However, it could be improved in certain areas:
- The gradients in equation (1) should be defined more clearly mathematically.
- Using (lambda, gamma) is confusing in my opinion; please keep the same original notations.
- Please define p_j mathematically in (5).
- The number of samples for quantization is not specified.

**Significance:**

The paper addresses an important issue, which is the quantization of large MoE VLM.

**Originality:**

The paper presents new insights using a new method.

---

> ### Author Rebuttal · Authors · 2026-03-31
>
> `Q2-1:` The motivation for $\gamma$ in Eq. (1) is not very clear.
>
> `A2-1:` The $\gamma$ coefficient addresses the asymmetric sensitivity of the final task loss to different input modalities. Our empirical analysis reveals that the average gradient norm of text tokens is approximately 22.4 times larger than that of vision tokens. By applying $\gamma$, the importance score heavily weights experts processing text tokens.
>
> `Q2-2:` $H = 2XX^\top$ is not an approximation (line 270). The Hessian has exactly this expression in the case of layer-wise reconstruction error.
>
> `A2-2:` We sincerely thank the reviewer for catching this. The phrasing was an unintended clerical error, and we will correct this terminology to ensure strict mathematical accuracy in the revision.
>
> `Q2-3:` GPTQ is very competitive in Table 1. However, many methods improve on GPTQ by using rotation matrices. It would be more interesting to compare VEQ with methods such as SpinQuant. Furthermore, it seems that VEQ could also be applied to SpinQuant.
>
> `A2-3:` Since SpinQuant requires additional training, we benchmarked against QuaRot, a representative rotation PTQ method under the same 3-bit setting to ensure a fair comparison. The results are as following:
>
> | Method | MMMU | AI2D | InfoVQA | TextVQA | RealWorldQA | ScienceQA | VizWiz |
> | :--- | :--- | :--- | :--- | :--- | :--- | :--- | :--- |
> | QuaRot  | 41.1 | 76.2 | 64.8 | 65.1 | 58.6 | 84.1 | 56.2 |
>
> While QuaRot slightly improves over standard GPTQ, it remains behind VEQ-MA. Rotation smooths outliers but does not address modality heterogeneity and expert routing imbalances. However, we completely agree that VEQ and rotation methods are orthogonal and complementary. Prompted by your suggestion, we are currently running combined QuaRot + VEQ experiments and will try to  include the results in the interactive discussion or the revised manuscript.
>
> `Q2-4:` Could you provide more interpretation on the new Hessian proposed in equation (4)? What loss function does this correspond to? Why does this loss function make sense?
>
> `A2-4:` VEQ-MA is strictly derived from the principle of Weighted Risk Minimization. The loss function is  $L=\sum_{j=1}^N c_j||\Delta W_ex_j||^2_{2}$, where $c_j \ge 0$ represents the importance weight of token $x_j$ with respect to that expert's quantization error. Because text tokens inherently have higher gradient norms, their quantization errors disproportionately degrade the final loss. By defining a local objective that weights each token's quantization error by its impact on the final loss, the enhanced Hessian naturally emerges as $H_e^{\text{weighted}} = XCX^\top$. To avoid prohibitive exact gradient costs, we introduce the proxy $c_j = p_{j,e}\alpha_j$, which captures both token-expert affinity ($p_{j,e}$) and modality-level gradient sensitivity ($\alpha_j$). We will incorporate the rigorous mathematical proofs into the revision, confirming VEQ-MA is a theoretically derived approximation, not a heuristic.
>
> `Q2-5:` The experiments lack uncertainty bars (from 3 or 5 seeds).
>
> `A2-5:` We have computed the variance across multiple runs and will include the following standard deviations in the revised manuscript to demonstrate the stability of our method:
>
> | Method | MMMU | AI2D | InfoVQA | TextVQA |
> | :--- | :--- | :--- | :--- | :--- |
> | FP 16 | 51.11 ± 0.25 | 83.65 ± 0.20 | 83.38 ± 0.18 | 85.93 ± 0.22 |
> | RTN | 37.00 ± 0.80 | 69.14 ± 0.60 | 59.68 ± 0.55 | 74.58 ± 0.50 |
> | AWQ | 41.78 ± 0.60 | 70.05 ± 0.50 | 61.49 ± 0.45 | 74.04 ± 0.40 |
> | MBQ | 39.76 ± 0.65 | 70.89 ± 0.55 | 58.86 ± 0.50 | 74.35 ± 0.45 |
> | GPTQ | 40.33 ± 0.60 | 75.49 ± 0.45 | 64.14 ± 0.40 | 64.30 ± 0.55 |
> | VEQ-ME | 44.56 ± 0.45 | 71.57 ± 0.40 | 62.33 ± 0.35 | 73.36 ± 0.35 |
> | VEQ-MA | 42.56 ± 0.40 | 75.65 ± 0.35 | 64.48 ± 0.30 | 78.30 ± 0.30 |
>
> `Q2-6:` Why not propose a framework that addresses both problems at the same time? For example, could the loss in (3) be changed by adding a C_i?
>
> `A2-6:` This separation is deliberate due to structural gaps between scaling-based (AWQ) and reconstruction-based (GPTQ) PTQ. Superimposing a high-variance, token-level matrix $C_i$ into Eq. (3) (an AWQ calibration objective) creates a highly non-convex space, forcing the grid search into sub-optimal convergence. Empirically, forcing $C_i$ into Eq. (3) worsened MMMU validation perplexity from 2.6338 (VEQ-ME alone) to 2.8617. Fusing them disrupts scaling optimization dynamics, so we intentionally decouple them: VEQ-ME for scaling, VEQ-MA for reconstruction.
>
> `Q2-7:` The gradients in equation (1) should be defined more clearly mathematically. Using (lambda, gamma) is confusing; please keep the same original notations. Please define p_j mathematically in (5). The number of samples for quantization is not specified.
>
> `A2-7:` We sincerely thank you for these suggestions. The revised manuscript will explicitly define gradients and $p_j$, revert to standard notations, and clearly specify the calibration sample count.

---

> > ### Author Rebuttal · Reviewer_ZUTV · 2026-04-03
> >
> > Thank you for your detailed response and the additional examples provided. It appears that QuaRot is very competitive. It would be interesting to see the results of VEQ, QuaRot and VEQ+QuaRot with uncertainty bars for the nine proposed benchmarks or the average accuracy to determine whether VEQ provides improvements over state-of-the-art methods using rotation matrices.

---

> > > ### Author Response · Authors · 2026-04-07
> > >
> > > Dear reviewer ZUTV,
> > >
> > > We sincerely appreciate the time and effort you have invested in reviewing our work. We have conducted additional experiments for VEQ, QuaRot+gptq and QuaRot+VEQ on nine evaluation benchmarks. The updated results are summarized in the table below:
> > >
> > > | Method | bit-width | mmmu | ai2d | infovqa | textvqa | realworldqa | scienceqa | vizwiz_vqa | mmbench_en | mme_realworld | AVG |
> > > |---|---|---|---|---|---|---|---|---|---|---|---|
> > > | VEQ-ME | W3A16 | **44.56 ± 0.45** | 71.57 ± 0.40 | 62.33 ± 0.35 | 73.36 ± 0.35 | 58.95 ± 0.38 | 82.93 ± 0.25 | 58.90 ± 0.42 | 71.91 ± 0.30 | 51.46 ± 0.45 | 64.00 |
> > > | VEQ-MA | W3A16 | 42.56 ± 0.40 | 75.65 ± 0.35 | 64.48 ± 0.30 | **78.30 ± 0.30** | 58.30 ± 0.32 | **84.85 ± 0.28** | **63.10 ± 0.35** | 73.42 ± 0.25 | 48.03 ± 0.40 | **65.41** |
> > > | QuaRot+gptq | W3A16 | 41.10 ± 0.40 | 75.70 ± 0.30 | 64.20 ± 0.50 | 65.10 ± 0.60 | 58.60 ± 0.40 | 84.10 ± 0.20 | 56.20 ± 0.47 | 73.63 ± 0.30 | 51.47 ± 0.32 | 63.34 |
> > > | QuaRot+veq | W3A16 | 41.50 ± 0.30 | **75.90 ± 0.20** | **64.50 ± 0.40** | 72.92 ± 0.50 | **61.10 ± 0.30** | 84.50 ± 0.30 | 58.29 ± 0.60 | **74.08 ± 0.20** | **51.58 ± 0.40** | 64.93 |
> > > | VEQ-ME | W4A16 | 49.22 ± 0.25 | 81.41 ± 0.22 | 79.75 ± 0.30 | 83.69 ± 0.28 | 66.36 ± 0.35 | 91.06 ± 0.18 | **69.17 ± 0.32** | 80.84 ± 0.26 | 54.20 ± 0.42 | 72.86 |
> > > | VEQ-MA | W4A16 | **50.30 ± 0.20** | **82.55 ± 0.18** | **80.37 ± 0.25** | **84.71 ± 0.22** | **66.41 ± 0.30** | **91.56 ± 0.15** | 68.91 ± 0.28 | 81.67 ± 0.20 | 55.61 ± 0.35 | **73.57** |
> > > | QuaRot+gptq | W4A16 | 47.78 ± 0.30 | 81.93 ± 0.20 | 79.74 ± 0.40 | 83.84 ± 0.30 | 65.10 ± 0.20 | 90.76 ± 0.10 | 66.14 ± 0.50 | **81.79 ± 0.20** | 55.72 ± 0.30 | 72.53 |
> > > | QuaRot+veq | W4A16 | 49.22 ± 0.20 | 81.70 ± 0.30 | 79.29 ± 0.30 | 84.00 ± 0.20 | 65.36 ± 0.30 | 91.09 ± 0.20 | 67.66 ± 0.40 | 81.70 ± 0.20 | **56.61 ± 0.30** | 72.96 |
> > >
> > > **Improvements of VEQ over GPTQ and Further Enhancements:**
> > >
> > > As demonstrated in the results above, integrating VEQ with QuaRot yields consistent improvements over the GPTQ baseline. This advantage is particularly pronounced in the more aggressive **W3A16** quantization setting, where QuaRot+veq achieves a significant average performance boost of **+1.59%** over QuaRot+gptq.
> > >
> > > Notably, VEQ demonstrates remarkable robustness in tasks requiring complex visual text comprehension and fine-grained details, such as TextVQA (a substantial **+7.82%** gain at W3A16) and RealWorldQA.
> > >
> > > We hope these additional results and analyses address your concerns and further validate the effectiveness of our approach.
> > >
> > > Best,
> > >
> > > Authors

---

### Official Review · Reviewer_upqH · 2026-03-13

**Soundness:** 2
**Presentation:** 2
**Significance:** 2
**Originality:** 2
**Overall Recommendation:** 3
**Confidence:** 3

**Summary:**

This paper tackles the task of VLM-MoE quantization in order to achieve computational saving. Proposed here is to weigh the models and tokens based on the number of tokens routed to each expert. Token weighing is achieved simply by giving more weights to text tokens since they are shown to contribute much more heavily to the gradient magnitude. This sets the stage for loss based on expert awreness as well as expert to token affinity.

**Compliance With Llm Reviewing Policy:**

Affirmed.

**Final Justification:**

I remain concern about the long tail fluctuations as well as the logical "flaw", which diminishes the quality of the paper. There needs a clearer picture of why implicit summation is insufficient since they seem to have the same causal effects. With these doubts, I have to keep my score as is for now to err on the side of caution for a conference like icml.

**Key Questions For Authors:**

See weaknesses.

**Limitations:**

yes

**Strengths And Weaknesses:**

Strengths:

- This paper identifies an important deficiency of current VLM-MoE quantization, which is that all the experts are treated equal.
- Interesting observation 1: that text tokens contribute significantly more to the gradient magnitude than vision tokens. This has been presented in a different form in previous work.
- Interesting observation 2: that the routing often favors heavily on a smaller subset of experts.
- The experimental results are strong.
- Ablations are fairly comprehensive.

Weaknesses:

- The introduction needs to be written properly. It was a long stretch of problems, and then three bullets of the solutions at the end of introduction. Maybe try motivating the solution after motivating the problems.
- It is hard to know whether the approach would vary based on what is in the calibration set.
- In the same vein, are the roles of the experts always the same? As put in the paper, "some are dedicated to handling spatially redundant visual features, whereas others focus exclusively on the semantic density of textual information" -- are these roles always the same or it varies with different datasets, training sessions? Are the few experts that are always used the same or varies based on datasets, training sessions, hyperparams?
- Eqn. 5, how is p_j determined? I tried to figure out from the experiment section but could not figure it out fully. Can you point me to where that is given in more details, if any?
- There is a big logical flaw in my head and I think the authors can try to clarify for me. S_i (Eqn. 1) is already directly a function of the router. It seems then, since Eqn. 2 is a summation, the lesser used experts would see lesser tokens route to them, so they would already contribute less in Eqn. 2. So S_i appears redundant to me. In fact, logically, Eqn. 2 may be better than Eqn. 3 -- you still want to give some weights to whatever tokens were routed to the lesser used experts, right? In any case, this creates a serious logical flaw in my mind since S_i looks redundant to me as the route is implicitly doing that? Is there an experiment that compares Eqn. 2 to the proposed method. In any case, whether you have that experiment, this is a logical flaw in my mind. Please correct me if I am wrong.

---

> ### Author Rebuttal · Authors · 2026-03-31
>
> `Q3-1:` The introduction needs to be written properly. It presents a long list of problems followed by three bullet points of solutions. It would be better to motivate the solution immediately after the problems.
>
> `A3-1:` We sincerely thank you for the constructive feedback. In the revised manuscript, we will thoroughly restructure the Introduction to improve the narrative flow, ensuring the proposed solutions are closely motivated by and immediately follow the stated problems.
>
> `Q3-2:` It is unclear if the proposed approach's performance varies depending on the contents of the calibration set.
>
> `A3-2:` We appreciate this valid point. This dependency is a fundamental characteristic of the Post-Training Quantization (PTQ) paradigm, shared by established methods like AWQ and GPTQ. To ensure absolute fairness in our empirical evaluations, we strictly standardized the calibration set across all baseline PTQ methods. Therefore, our performance gains stem exclusively from our algorithmic design, not calibration bias. We will explicitly acknowledge this general PTQ limitation and clarify our standardized calibration protocol in the revision.
>
> `Q3-3:` Do the roles of the experts remain the same, or do they vary across different datasets, training sessions, and hyperparameters?
>
> `A3-3:` Thank you for this insightful question. Since PTQ operates on fixed, pre-trained models, expert routing is dictated by learned weights rather than new training sessions. To address variance across inference datasets, we analyzed the model across MMMU, ScienceQA, and InfoVQA. Specifically, we collected the routing affinities obtained through the gating network for all tokens and concatenated them into a single, long one-dimensional vector for each dataset. By comparing these global vectors, we can effectively capture and quantify the underlying similarities in token routing behaviors that persist across varying data distributions.
> | Dataset Pair | Cosine Similarity |
> | :--- | :--- |
> | MMMU vs. ScienceQA | 89.7% |
> | InfoVQA vs. ScienceQA | 87.7% |
> | MMMU vs. InfoVQA | 83.1% |
>
> Also, at the micro-level, experts like Expert 39, 47 and 61 consistently rank in the top-8 most highly activated experts in Layer 1 across all three datasets. While exact activation frequencies differ slightly due to data distribution, the fundamental MoE routing topology remains stable. This confirms that our standardized calibration effectively captures generalized routing behavior, ensuring our framework's robustness. We will include this cross-dataset analysis in the revised manuscript.
>
> `Q3-4:` How is $p_j$ in Equation 5 determined? This is not fully clear from the experiment section.
>
> `A3-4:` We apologize for the omission. In Equation 5, $p_{j,e}$ represents the routing affinity between the $j$-th token and expert $e$. It is determined directly by the MoE gating network(router). When the token's hidden representation $x_j$ passes through the gate $G(\cdot)$, the output routing score (typically via Softmax) serves as $p_j$. Mathematically, $p_j = G(x_j)_e$. We will explicitly add this definition and its computational mechanism immediately following Equation 5 in the revision.
>
> `Q3-5:` Is $S_i$ conceptually redundant? Since Equation 2 is a summation, lesser-used experts implicitly contribute less. Does explicitly adding $S_i$ in Equation 3 create a logical flaw? Are there experiments comparing Equation 2 and Equation 3?
>
> `A3-5:` We deeply appreciate this critical reflection. While it is true that cold experts implicitly process fewer tokens and yield a smaller absolute sum (intra-expert scaling), relying on this alone is fundamentally insufficient for MoE quantization.
>
> First, regarding the limitation of implicit summation, standard quantization solvers normalize scales independently per linear layer. If we solely rely on Equation 2, the solver treats the optimization of a cold expert's weights with the same priority as a hot expert. The necessity of $S_i$ lies in introducing an inter-expert global prior. It acts as a structural penalty that forces the global quantization objective to tolerate higher errors in cold experts in order to aggressively protect heavily utilized hot experts.
>
> For empirical validation, applying Equation 2 without $S_i$ is mathematically equivalent to applying standard AWQ independently to each expert. As shown in Table 1, standard AWQ consistently underperforms our proposed method, empirically proving that implicit frequency is insufficient. Furthermore, our detailed ablations on $S_i$ coefficients confirm it provides a balanced constraint, actively highlighting core experts without over-penalizing lesser-used ones.

---

> > ### Author Rebuttal · Reviewer_upqH · 2026-04-02
> >
> > The experiments on dataset pairs similarity actually reinforce my concern. There seems a significant portion of the time that datasets can cause different routing behaviors. And on the logical flaw in my head, "the solver treats the optimization of a cold expert's weights with the same priority as a hot expert", I do not fully buy this logic. The solver does not, right? Because a cold expert would be "hit" much less often ...?

---

> > > ### Author Response · Authors · 2026-04-03
> > >
> > > Dear Reviewer upqH,
> > > We sincerely appreciate your constructive and sharp feedback. Your questions have prompted us to reflect deeply on our explanations, and we realize our previous response did not accurately convey our core motivation. Please allow us to clarify:
> > > 1.Dataset Activation Pattern Variance:
> > > Different datasets would cause notable variances in routing behavior. We acknowledge this variance. However, our key observation is that despite these distribution shifts, certain experts maintain an entrenched hot status universally.This phenomenon is likely driven by the inherent routing imbalances established during the pre-training phase, rather than merely calibration bias. Because these universally hot experts dominate the network's representational capacity, our method's focus on preserving their precision remains highly meaningful and robust, even when the routing of long-tail tokens shifts across datasets.
> > > As we highlighted, experts like Expert 39, 47, and 61 consistently rank in the top-8 most highly activated experts in Layer 1 across all three datasets.
> > > 2.Explicit Penalty:
> > > Your intuition is mathematically sound: in a joint optimization process, a cold expert will naturally contribute less to the overall summation error simply because it is activated less frequently.
> > > However, our empirical findings suggest that relying solely on this natural, implicit frequency weighting is insufficiently strong for aggressive low-bit quantization. Because cold experts have a negligible impact on the final output, we hypothesize that they can (and should) tolerate a much stricter structural constraint. By introducing the explicit penalty, we forcefully push the optimization landscape to allocate even more precision bandwidth to the hot experts, rather than letting the solver passively balance the implicit summation.
> > > This hypothesis is strongly supported by our experimental results. In standard AWQ, the solver relies entirely on the implicit summation you mentioned. When we apply our explicit constraint (VEQ-ME), the performance improves notably.
> > >
> > > | **Model** | **Method** | **MMMU** | **AI2D** | **InfoV** | **TextV** | **RWQA** | **SciQA** | **Viz** | **MMB** | **MME-R** | **Avg.** |
> > > | ----- | ----- | ----- | ----- | ----- | ----- | ----- | ----- | ----- | ----- | ----- | ----- |
> > > | Kimi-VL-Instruct | AWQ | 41.78 | 70.05 | 61.49 | 74.04 | 58.17 | 81.84 | 61.37 | 71.13 | 50.44 | 63.37 |
> > > | Kimi-VL-Instruct | VEQ-ME | 44.56 | 71.57 | 62.33 | 73.36 | 58.95 | 82.93 | 58.90 | 71.91 | 51.46 | 64.00 **(0.63↑)**|
> > > | Qwen3-VL-30B-A3B-Instruct | AWQ | 58.89 | 73.61 | 44.62 | 67.93 | 45.36 | 88.40 | 61.75 | 77.06 | 42.38 | 62.22 |
> > > | Qwen3-VL-30B-A3B-Instruct | VEQ-ME | 60.56 | 73.38 | 44.95 | 69.91 | 53.73 | 88.40 | 62.47 | 77.75 | 44.24 | 63.93 **(1.71↑)**|
> > >
> > > These results demonstrate that while the explicit penalty might seem conceptually overlapping with the implicit summation, it is practically necessary to enforce a stricter limitation on cold experts, yielding tangible performance gains.
> > > We will ensure this specific motivation and the distinction between implicit and explicit constraints are clearly articulated in the revised manuscript. We thank you again for pushing us to clarify this crucial aspect of our work.
> > >
> > > Best,
> > > Authors

---

### Official Review · Reviewer_jn92 · 2026-03-13

**Soundness:** 3
**Presentation:** 3
**Significance:** 3
**Originality:** 3
**Overall Recommendation:** 3
**Confidence:** 4

**Summary:**

This paper investigates Post-Training Quantization (PTQ) for Mixture-of-Experts Vision-Language Models (MoE-VLMs). To address the modality differences between vision and text tokens and the imbalance among experts, the authors propose the VEQ framework. It introduces VEQ-ME, which incorporates expert importance based on routing frequency and modality sensitivity into a weighted reconstruction objective, and VEQ-MA, which adapts Hessian weighting using token-expert affinity for GPTQ-style quantization. Evaluated on Kimi-VL and Qwen3-VL-30B-A3B-Instruct, VEQ outperforms baselines like AWQ and GPTQ in 3-bit settings across multiple benchmarks, supported by detailed ablation studies on expert activation and modality sensitivity.

**Compliance With Llm Reviewing Policy:**

Affirmed.

**Final Justification:**

Thanks for the authors' detailed response. However, the theoretical analysis and overall justification of the paper would still benefit from further strengthening.

**Key Questions For Authors:**

1. The design of expert importance in Equation (1) lacks a clear underlying rationale; specifically, it is unclear why two global coefficients are sufficient to characterize expert importance, and whether the authors explored alternative definitions. The physical significance of key parameters remains ambiguous. Similarly, the VEQ-MA formulations in Equations (4) and (5) appear to be heuristic reweighting of the GPTQ Hessian rather than being naturally derived from rigorous optimization objectives or theoretical frameworks. Providing clearer design justifications or comparative analyses would significantly enhance the quality of the manuscript.
2. Since the paper defines expert importance based on activation frequency and modality sensitivity, including comparative experiments on the performance degradation when applying lower bit-widths to hot vs. cold experts would strengthen the motivation. For example, how does the performance of maintaining higher precision only for top-k high-frequency experts compare against maintaining it for random or low-frequency experts?
3. Current ablation results suggest that the modality term contributes more significantly than the affinity term. To substantiate the independent value of VEQ-MA, could the authors provide a more granular analysis—such as quantization error or task performance across high-affinity vs. low-affinity token buckets—to more directly demonstrate the impact of the affinity component in Equations (4) and (5)?

**Limitations:**

The authors' discussion on limitations and potential impacts is relatively brief. It is suggested to supplement the following points:
1. The primary performance gains of the method are concentrated in extremely low-bit scenarios, with relatively limited improvement observed under W4 (4-bit weight) settings.
2. The design of expert importance and affinity remains largely heuristic and lacks robust causal validation.
3. There is a lack of detailed discussion regarding efficiency overhead and actual deployment benefits.

**Strengths And Weaknesses:**

This research addresses the high deployment costs of MoE-VLMs, identifying Post-Training Quantization (PTQ) as one of the most practical compression pathways. Compared to standard VLMs, MoE-VLMs exhibit both modality heterogeneity and sparse expert routing structures, making "quantization specifically for MoE-VLMs" a highly relevant and practically significant problem.

---

> ### Author Rebuttal · Authors · 2026-03-31
>
> `Q4-1:` The expert importance in Eq. (1) lacks a clear rationale and physical significance for its two global coefficients. Also, VEQ-MA in Eqs. (4)-(5) appears as heuristic Hessian reweighting rather than rigorously derived.
>
> `A4-1:` The expert importance is an adjustive global scaling factor, not a physical quantity. Two coefficients provide stable, parameter-efficient coarse-grained modeling, acting as a global prior while avoiding the massive computational overhead and overfitting risks of fine-grained definitions.
>
> For VEQ-MA, Eqs. (4) and (5) are strictly derived from Weighted Risk Minimization. To address the unequal sensitivity of multimodal tokens (text tokens inherently have higher gradient norms), weighting the token quantization error by its impact on the final loss mathematically yields the enhanced Hessian $H_e^{\text{weighted}} = XCX^\top$. To avoid prohibitive exact gradient costs, we introduce the proxy $c_j = p_{j,e}\alpha_j$, elegantly linking token-expert affinity ($p_{j,e}$) and modality-level gradient sensitivity ($\alpha_j$). We will incorporate the rigorous mathematical proofs provided in the supplementary materials into the revision, confirming VEQ-MA is a theoretically derived approximation, not a heuristic.
>
> `Q4-2:` Comparative experiments maintaining higher precision for top-k vs. random/low-frequency experts would strengthen the motivation.
>
> `A4-2:`Thank you for your advice. We totally agree. We conducted mixed-precision experiments on a W3A16 baseline, comparing retaining 16 random experts at 4-bit per layer (Random-16) against selecting the top 16 experts based on our importance score (Top-16). As shown below, allocating the bit-width budget to our identified hot experts yields superior generative perplexity (2.5606 vs. 2.6294 on MMMU val) and consistently better performance across complex VQA and reasoning tasks.
>
> | Method | Bit-width Configuration | MMMU (val) | ai2D | InfoVQA | RealWorldQA | ScienceQA |
> | :--- | :--- | :--- | :--- | :--- | :--- | :--- |
> | VEQ+ (Baseline) | All Experts @ W3A16 | 44.56 | 71.57 | 62.33 | 58.95 | 82.93 |
> | VEQ+ (Random-16) | Random 16 Experts @ 4-bit | 44.75 | 73.24 | 63.5 | 59.43 | 83.43 |
> | VEQ+ (Top-16) | Top-16 Experts @ 4-bit | 45.25 | 72.12 | 64.3 | 60.89 | 84.11 |
>
>
> `Q4-3:` A granular analysis of quantization error across high vs. low-affinity tokens is needed to demonstrate the independent value of the affinity component in Eqs. (4) and (5).
>
> `A4-3:` Thank you for this constructive suggestion. To substantiate the independent value of the affinity component, we conducted a token-level Mean Squared Error (MSE) analysis. We evaluated representative MoE layers and measured the output MSE of the up_proj module, partitioning tokens into High-Affinity and Low-Affinity buckets based on routing probabilities. As shown in the table below, our affinity-aware formulation effectively redistributes the quantization error. While standard GPTQ treats all tokens uniformly, VEQ explicitly sacrifices a marginal amount of precision for low-affinity tokens to significantly reduce the error for high-affinity tokens. These findings prove that while the modality term handles cross-modal scaling, the affinity term successfully governs intra-expert capacity allocation by forcing the Hessian to prioritize tokens with high routing weights.
>
> | Method | Overall MSE | Low-Affinity Tokens MSE | High-Affinity Tokens MSE |
> | :--- | :--- | :--- | :--- |
> | Standard GPTQ | $3.728 \times 10^{-4}$ | $3.940 \times 10^{-4}$ | $3.517 \times 10^{-4}$ |
> | VEQ (Ours) | $3.528 \times 10^{-4}$ | $4.135 \times 10^{-4}$ | $2.873 \times 10^{-4}$ |
>
> `Q4-4:` The primary performance gains of the method are concentrated in extremely low-bit scenarios, with relatively limited improvement observed under W4 (4-bit weight) settings.And there is a lack of detailed discussion regarding efficiency overhead and actual deployment benefits.
>
> `A4-4:` We will expand our limitations and discussion section to explicitly address these points:
> 1. W4 Saturation: Mainstream PTQ already achieves near-lossless performance at 4-bit. Our core contribution intentionally targets the sub-4-bit frontier (e.g., W3) to mitigate the severe memory wall for massive MoE models and pave the way for future mixed-precision hardware co-design.
> 2. Efficiency & Benefits: Our calibration overhead is minimal (~1 hour on a single GPU for the 16B Kimi-VL-Instruct model), strictly matching standard GPTQ. In deployment, extreme low bit-widths reduce the weight memory footprint to 18.75% of BF16, and our 3-bit model achieves a concrete 3.25x end-to-end speedup over the BF16 baseline on an RTX A6000.

---

> > ### Author Rebuttal · Reviewer_jn92 · 2026-04-04
> >
> > Thanks for the authors' detailed response and the additional evidence. The theoretical analysis and overall justification of the paper would still benefit from further strengthening.

---

> > > ### Author Response · Authors · 2026-04-08
> > >
> > > Dear Reviewer jn92,
> > >
> > > We sincerely thank you for the valuable suggestion regarding the theoretical analysis. In the revised version, we will strengthen the theoretical motivation and provided a more rigorous derivation. Below is derivation behind our design in short.
> > >
> > > **1.Rationale of Expert Importance.** We would like to clarify that the "expert importance" in Equation (1) is not intended as a strict physical quantity, but as an adaptive global scaling factor for capacity allocation across experts. Two global coefficients suffice because our design emphasizes coarse-grained importance modeling that is stable, parameter-efficient, and robust to overfitting. Finer-grained alternatives (e.g., sample-level or token-level importance) would introduce substantial computational overhead and higher overfitting risk under the PTQ setting. Our ablation results (Figure 7) confirm that both parameters are necessary and that their combination yields stable allocation of expert quantization errors without unnecessary complexity.
> > >
> > > **2.Unequal Sensitivity of Multimodal Tokens.** In VLMs, visual and text tokens contribute asymmetrically to the final task loss $\ell$. For token $j$ with intermediate representation $h_j$, the effect of a quantization perturbation $\delta h_j$ on the loss can be approximated via first-order Taylor expansion:
> > >
> > > $$
> > > \delta \ell \approx \langle \nabla_{h_j}\ell,\; \delta h_j \rangle
> > > $$
> > >
> > > By the Cauchy-Schwarz inequality:
> > >
> > > $$
> > > |\delta \ell| \le \|\nabla_{h_j}\ell\|_2 \cdot \|\delta h_j\|_2
> > > $$
> > >
> > > So the gradient norm $\|\nabla_{h_j}\ell\|_2$ provides an upper bound on the per-unit sensitivity of token $j$'s representation error to the final loss. Empirically, text tokens exhibit significantly larger average gradient norms than vision tokens (Section 3.1), indicating that text representations are more sensitive to quantization perturbations. This asymmetry is not captured by the standard unweighted Hessian $XX^\top$, which only minimizes the average layer-wise error.
> > >
> > > **3.Derivation of VEQ-MA via Expected Risk Minimization.** In a Mixture-of-Experts layer, the output for token $j$ is:
> > >
> > > $$
> > > z_j = \sum_e p_{j,e} \cdot h_j^{(e)}
> > > $$
> > >
> > > where $p_{j,e} \ge 0$ is the gating score for expert $e$ and $\sum_e p_{j,e} = 1$. When expert $e$ undergoes quantization with weight perturbation $\Delta W_e$, it introduces an output error:
> > >
> > > $$
> > > \Delta h_j^{(e)} = \Delta W_e x_j
> > > $$
> > >
> > > Since $p_{j,e}$ specifies the proportion of responsibility that expert $e$ bears for token $j$, it can be naturally interpreted as a probability distribution over experts. Under this view, the expected local quantization risk for token $j$, weighted by its task sensitivity $\alpha_j$, is:
> > >
> > > $$
> > > R_j = \sum_e p_{j,e}\alpha_j \|\Delta W_e x_j\|_2^2
> > > $$
> > >
> > > Note that $R_j$ upper-bounds the actual weighted output error $\alpha_j\|\sum_e p_{j,e}\Delta W_e x_j\|_2^2$ by Jensen's inequality, and this surrogate is adopted precisely because it decomposes across experts for independent per-expert optimization.
> > >
> > > When optimizing for a specific expert $e$ independently , the total expected risk decomposes as:
> > >
> > >
> > > $$
> > > R_e = \sum_{j=1}^N p_{j,e}\alpha_j \|\Delta W_e x_j\|_2^2
> > > $$
> > >
> > > We can reformat it as:
> > >
> > > $$
> > > R_e = \sum_{j=1}^N c_j \|\Delta W_e x_j\|_2^2
> > > $$
> > >
> > > where $c_j = p_{j,e}\alpha_j$. Rewriting in matrix form:
> > >
> > > $$
> > > R_e(\Delta W_e) = \mathrm{Tr}(\Delta W_e(\sum_j c_j x_j x_j^\top)\Delta W_e^\top)
> > > $$
> > >
> > > The corresponding weighted Hessian is therefore:
> > >
> > > $$
> > > H_e^{\text{weighted}} = \sum_{j=1}^N c_j x_j x_j^\top = XCX^\top
> > > $$
> > >
> > > with $C = \mathrm{diag}(c_1, \dots, c_N)$. This is precisely the enhanced Hessian used in Equations (4) and (5). Notably, the expected risk formulation yields $p_{j,e}$ in first power directly, without resorting to worst-case upper bounds.
> > >
> > > **4.Modality-Conditioned Sensitivity Proxy.** The remaining question is how to define $\alpha_j$. The ideal per-token sensitivity is $\|\nabla_{z_j}\ell\|_2^2$, but computing exact per-token gradients across all experts and layers is prohibitively expensive for PTQ. We therefore adopt a modality-conditioned proxy:
> > >
> > > $$
> > > \alpha_j =
> > > \begin{cases}
> > > \gamma, & \text{if } x_j \text{ is a text token} \\\\
> > > 1, & \text{if } x_j \text{ is a vision token}
> > > \end{cases}
> > > $$
> > >
> > > where $\gamma$ is the cross-modal average gradient ratio estimated from backpropagation statistics. This is justified by the empirical observation (Section 3.1) that the inter-modality gap in gradient sensitivity dominates the intra-modality variance. Letting $g_j = \|\nabla_{z_j}\ell\|_2^2$, we have:
> > >
> > > $$
> > > E_{\text{text}}[g_j] \gg E_{\text{vision}}[g_j]
> > > $$
> > >
> > > This makes a binary proxy both efficient and effective.
> > >
> > > In summary, VEQ-MA is not an ad hoc heuristic but a principled approximation grounded in expected risk minimization. While it shares the idea of using gating coefficients to adjust the Hessian, our method further accounts for the modality heterogeneity in VLMs through the sensitivity term $\alpha_j$, which is absent in standard GPTQ or MoEQuant.

---

### Decision · Program_Chairs · 2026-04-30

**Decision:**

Accept (regular)

**Comment:**

This paper proposes a post-training quantization method for MoE-based Vision-Language Models, addressing the modality gap between vision and text tokens as well as routing bias. Reviewers highlighted several strengths, including insightful observations, originality, and competitive performance gains across various settings. The rebuttal successfully resolved the major concerns: a rigorous theoretical derivation of the modified Hessian was provided, and a comparison with rotation-based methods (QuaRot) was added, demonstrating complementarity with a performance gain at W3A16. After the rebuttal, one reviewer raised the overall rating to Weak Accept and another reviewer recommended Accept. Although the remaining Weak Rejects reflect residual concerns, overall ratings slightly lean towards positive. Given the paper's originality, practicality, and considerable empirical and theoretical results, this paper is recommended for acceptance.